mathematical modelling/computer modelling and simulation/computational biology

hybrid model, spatio-temporal dynamics, agent-based model, viral kinetics, SARS-CoV-2, influenza

**Author for correspondence:**
Sadegh Marzban
e-mail: sadegh.marzban@math.u-szeged.hu

# A hybrid PDE–ABM model for viral dynamics with application to SARS-CoV-2 and influenza

Sadegh Marzban, Renji Han, Nóra Juhász and Gergely Röst

Bolyai Institute, University of Szeged, Szeged 6720, Hungary

 SM, 0000-0002-9435-017X; RH, 0000-0001-5899-9943;
NJ, 0000-0001-6560-7581; GR, 0000-0001-9476-3284

We propose a hybrid partial differential equation–agent-based (PDE–ABM) model to describe the spatio-temporal viral dynamics in a cell population. The virus concentration is considered as a continuous variable and virus movement is modelled by diffusion, while changes in the states of cells (i.e. healthy, infected, dead) are represented by a stochastic ABM. The two subsystems are intertwined: the probability of an agent getting infected in the ABM depends on the local viral concentration, and the source term of viral production in the PDE is determined by the cells that are infected. We develop a computational tool that allows us to study the hybrid system and the generated spatial patterns in detail. We systematically compare the outputs with a classical ODE system of viral dynamics, and find that the ODE model is a good approximation only if the diffusion coefficient is large. We demonstrate that the model is able to predict SARS-CoV-2 infection dynamics, and replicate the output of *in vitro* experiments. Applying the model to influenza as well, we can gain insight into why the outcomes of these two infections are different.

## 1. Introduction

Mathematical models have been powerful tools in tackling the challenges posed by the appearance of the COVID-19 disease caused by the novel SARS-CoV-2 coronavirus. In the ongoing and virtually unprecedented pandemic, these mathematical models are invaluable as they are able to provide insights or predictions based on mathematical analysis and computer simulations. This means that their results are obtained at an ideally low cost even in complex situations where real-life and real-time experiments to obtain these same results would be

either extremely risky or simply not possible. There is a large variety of both large- and small-scale mathematical models related to COVID-19. The present article is dedicated to cellular level investigations—we examine virus dynamical phenomena such as in-host viral transmission between individual cells. Specifically, we use a hybrid mathematical approach for our study, focusing on SARS-CoV-2 and influenza infections.

Throughout the investigation of the above-mentioned hybrid model, we need a simple and already comprehensively understood model that we can use as a reference system—for this purpose, we will consider a specific version of an ODE prototype model, the May–Nowak system (for an extensive overview, see [1]; while for a concrete application, see [2]). This classical model is given by

$$\begin{cases} H_t = -\beta HV, & t > 0, \\ I_t = \beta HV - \delta I, & t > 0, \\ V_t = pI - cV, & t > 0, \end{cases}$$

where the $H = H(t)$, $I = I(t)$ and $V = V(t)$ functions respectively correspond to the number of healthy cells, infected cells and virus particles, and they depend on time only. This model plays a crucial part in our article in several ways: (a) in the development process of the more sophisticated hybrid model, we use the ODE model to compare and match the respective solutions, (b) we exploit the fact that there is more information available on the exact parameters of this classical model and use their converted version in the hybrid system, and finally (c) we can demonstrate both the advantages and challenges offered by the relatively new hybrid system in comparison to a well-known model.

The above ODE model is widely used to describe in-host viral dynamics. While this method proves to be extremely useful, the quality of information it can provide is naturally limited since it uses time as its single independent variable, and cannot capture spatial effects. Spatio-temporal phenomena are ubiquitous in biological applications, and often we simply cannot afford to leave out such an important factor from our models: the most obvious example for this is virus diffusion itself, which makes it possible for viruses to physically reach susceptible cells. One of the most powerful tools that does capture these crucial spatial mechanisms is the rich analytical theory of partial differential equations—PDE models are often used to investigate complex systems in viral infection dynamics; see [3] just to mention one SARS-CoV-2-related example.

Agent-based modelling is another (and in fact quite different) approach to describe complex space–time dynamics in viral spread. The cornerstone idea of this method is to define a discrete heterogeneous state space where the so-called agents or elements have collective interactions with each other and they change their states correspondingly [4]. The agent-based model (ABM) concept has several important applications within the field of virus dynamics; for example, in [5] the authors apply an ABM to simulate influenza interactions at the host level.

As seen above, different frameworks—including ABMs and PDE systems—have been applied to construct viral dynamics models. An ABM is a great tool to include randomness and natural variability into the system, while PDEs are faster to evaluate numerically: by forming bridges between these two modelling strategies we arrive at a hybrid system, the so-called hybrid PDE–ABM model, which unites the different advantages of these methods (see [6]). For the purpose of investigating the in-host viral spread of SARS-CoV-2 and influenza, our hybrid model is chosen and tailored by carefully considering certain dimension-related aspects of the problem: virus particles are several orders of magnitude smaller than epithelial cells [7–9], and as a consequence, it is natural to capture virus concentration by a continuous function, while cells are modelled in a straightforward way as discrete entities in space. In other words, we obtain our hybrid model by merging (a) a PDE representing virus concentration and (b) an ABM describing host cells and their three possible states (healthy, infected and dead). We complete the hybrid PDE–ABM model by formulating a meaningful connection between the separate parts, i.e. by giving definition to all the considerable interactions and feedback processes that take place between the model's discrete and continuous parts, such as for example cells responding to their environment's virus concentration level. We highlight that our implementation of the proposed hybrid system is based on a free and open source software package, HAL (Hybrid Automata Library) [10].

The present article's novelty consists both in its approach and results. While PDE–ABM hybrid models have existed before HAL [10], the latter is primarily focused to serve cancer-related research—we have adapted, configured, and applied this tool to allow investigations in a completely different field of application: host-level virus infections. Our methodology and results demonstrate that both our model and its implementation technique provide a suitable framework to investigate problems in virology and immunology. We emphasize the importance of monitoring virus propagation in both

space and time—we show that simpler models that disregard spatial phenomena correspondingly lose predictive precision due to ignoring spatial diffusion. The numerical simulations we perform using the hybrid system as a foundation give spatially explicit information regarding cellular-level virus spread for SARS-CoV-2 in a high-resolution state space, which is one of the main achievements of our article. The successful, computer-simulated replication of real *in vitro* experiments regarding SARS-CoV-2 propagation is also new to our knowledge. Finally, we highlight that our results allow us to gain insight into why the respective outcomes of influenza and SARS-CoV-2 infections are different.

# 2. Methods

The purpose of this section is to give a detailed description of the two main systems we use to model virus spread. We begin by defining the hybrid PDE–ABM model and introducing its variables, then we briefly outline a classical ODE model—the latter can be viewed as a simpler mean-field model that approximates the more complex hybrid framework by averaging over spatial variables. Once all the fundamental features of both systems are defined, we continue this section by expressing the connection between the corresponding model parameters, and conclude with the numerical implementation of the hybrid system.

## 2.1. The hybrid PDE–ABM model

As described in the Introduction, the main hybrid system is constructed via forming bridges between two important and rather different modelling techniques: we merge a discrete ABM and a continuous PDE, and we create a meaningful connection between them by carefully designing their interactions with each other. We highlight the fact that the model we obtain in this way is defined in both space and time.

We begin by setting the notation for the domain we construct our model upon: let $\Omega$ be the mathematical representation of the part of lung tissue, or in the case of an *in vitro* experiment, the relevant area of the investigation we are considering. Now we are ready to introduce the discrete part of our hybrid system.

One of the most important modelling decisions in the first part of the model construction is approaching epithelial cells as discrete agents. In more detail, we define a two-dimensional ABM state space by introducing a lattice of $k_1 \times k_2$ agents representing epithelial cells: naturally, $k_1, k_2 \in \mathbb{N}$, and cells are identified via the $(i, j)$ indices (corresponding to the respective agent's place in the grid), where $(i, j) \in \mathcal{J} = \{(i, j) | 1 \leq i \leq k_1, \ 1 \leq j \leq k_2\}$. By introducing the $\Omega_{i,j}$ notation for the open set occupied by the $(i, j)$th cell, we obtain that $\bar{\Omega} = \bigcup_{(i,j) \in \mathcal{J}} \bar{\Omega}_{i,j}$.

Regarding the states of the ABM space's agents, each agent can have three possible states in our approach. This is formally grasped by the following state function, which represents the concept that a cell is either *healthy*, *infected* or *dead*:

$$s_{i,j}(t) = \begin{cases} H, & \text{if the } (i, j)\text{th cell is healthy at time } t \\ I, & \text{if the } (i, j)\text{th cell is infected at time } t \\ D, & \text{if the } (i, j)\text{th cell is dead at time } t. \end{cases}$$

In terms of state dynamics, we use the following assumptions:

— for simplicity, we do not account for cell division or cell birth taking place during the time frame of the infection;
— the only reason for cell death is viral infection itself, i.e. death related to any other natural cause is ignored;
— all healthy cells are susceptible target cells;
— once a cell gets infected, it is not possible for it to become healthy again;
— the *healthy → infected* state change: a healthy cell may become infected once the virus has reached the given cell; moreover, infection is randomized and it occurs with a probability of $P_I$, where we highlight that $P_I$ increases linearly with the virus concentration in the given cell (for more details see §2.3 on $P_I$'s relationship to other parameters and §2.4 on implementation);
— the *infected → dead* state change: an infected cell dies with a probability $P_D$, but similarly to infection, death too is approached from a stochastic viewpoint.

The stochastic element in the above state changes is of key importance. The inclusion of natural randomness makes the complex PDE–ABM hybrid model not only more realistic than the relatively simple ODE model, but also compared to a pure PDE system.

We note that apart from following the state changes of all cells on an individual level, we also introduce three additional system variables as functions taking discrete (in fact, natural) values: $H(t)$, $I(t)$ and $D(t)$, respectively, denote the total number of healthy, infected and dead cells in the two-dimensional ABM state space.

Now we are ready to consider the second part of our hybrid model. Let us begin by observing that the size of an epithelial cell is relatively significant [8,9] and hence indeed it was a natural idea above to define cells as separate entities and follow their states on an individual level (instead of exclusively considering the total number of healthy, infected and dead cells as continuous functions for example). One of the cornerstone observations in the second part of the model is that viruses, on the other hand, are several orders of magnitude smaller compared to epithelial cells [7]—we incorporate this simple but crucial biological fact into our system by modelling virus concentration as a continuous function. Hence, in the second part of the model, we approach virus concentration $V^h = V^h(t, x, y)$ as a variable that is continuous in both space and time, and as such, we capture it by means of a partial differential equation:

$$\frac{\partial V^h(t, x, y)}{\partial t} = D_V \Delta V^h - \mu_V \cdot V^h(t, x, y) + \sum_{(i,j) \in \mathcal{J}} g_{i,j}(t, x, y), \quad t > 0, \ (x, y) \in \Omega, \tag{2.1}$$

where $D_V$ denotes the virus diffusion coefficient, $\mu_V$ is a constant ratio representing virus removal, while $g_{i,j}$ stands for the viral source term (the latter is assumed to be continuous) for the $(i, j)$th cell. The above equation essentially grasps the basic concept that viruses spread across the domain primarily via diffusion (convective flows are ignored in this model), the immune system removes viruses in a constant ratio, while new virus particles are generated by infected cells in a process described by the $g_{i,j}$ functions

$$g_{i,j}(t, x, y) = \begin{cases} 0, & \text{if } s_{i,j}(t) = H \text{ and } (x, y) \in \Omega_{i,j} \\ f_{i,j}(t, x, y), & \text{if } s_{i,j}(t) = I \text{ and } (x, y) \in \Omega_{i,j} \\ 0, & \text{if } s_{i,j}(t) = D \text{ and } (x, y) \in \Omega_{i,j} \\ 0, & \text{if } (x, y) \notin \Omega_{i,j}. \end{cases} \tag{2.2}$$

We point out that in the above formula, we do not specify any particular exact $f_{i,j}(t, x, y)$ form for the viral source term $g_{i,j}(t, x, y)$ for the case when $s_{i,j}(t) = I$ and $(x, y) \in \Omega_{i,j}$ (i.e. when the $(i, j)$th cell is infected). As noted in [11], very little is known about the shape, duration and magnitude of viral burst, and hence we allow any reasonable and smooth $f_{i,j}(t, x, y)$ function here that meets the following two criteria. Firstly, any concrete choice for $g_{i,j}$ needs to represent the general fact that the $(i, j)$th epithelial cell starts secreting virions at some point after it becomes infected. Thus, naturally, $g_{i,j}$ takes positive values at least in some subset of $\Omega_{i,j}$ at some point after the event of infection (but of course, as formulated above, outside of this given cell $g_{i,j}$ is zero). Secondly, the $g_{i,j}$ function needs to be defined in a way so that the well-posedness of system (2.1) is guaranteed: specifically, any concrete definition of the viral source term has to be Hölder continuous with Hölder exponent $\alpha \in (0, 1)$, i.e. we assume $g_{i,j}(t, x, y) \in C^{(\alpha/2),\alpha}((0, \infty) \times \bar{\Omega})$ for any $t > 0, (x, y) \in \bar{\Omega}$ and any $(i, j) \in \mathcal{J}$. The latter ensures the global existence of the solution of system (2.1) (see appendix B). As an example, we can consider a definition where we use the modified version of a constant rate release within the $(i, j)$th cell. As for the details and changes related to the construction of $g_{i,j}$ in the implementation process, see §2.4.

The PDE–ABM hybrid model cannot be complete without explicitly taking into account the meaningful interactions that each part of the system has on the other, hence we briefly highlight these connections again. The continuous viral part affects the agent-based subsystem through the *healthy →
infected* state change described in the ABM section above. On the other hand, the discrete ABM part makes a difference within the continuous viral equation thanks to the $g_{i,j}$ source functions (representing the fact that once a cell gets infected, at some point it starts spreading the virus).

## 2.2. The ODE model

In this subsection, we consider a simpler viral dynamics system which will serve as a reference model for the hybrid framework in our epidemiological investigations. We consider the following—only time-

dependent—ODE system:

$$
\begin{cases}
\dfrac{dH(t)}{dt} = -\beta H(t)V(t), \; t > 0, \\[2mm]
\dfrac{dI(t)}{dt} = \beta H(t)V(t) - \delta I(t), \; t > 0, \\[2mm]
\dfrac{dV(t)}{dt} = pI(t) - cV(t), \; t > 0, \\[2mm]
\dfrac{dD(t)}{dt} = \delta I(t), \; t > 0, \\[2mm]
H(0) = H_0 \geq 0, \; I(0) = I_0 \geq 0, \; V(0) = V_0 \geq 0, \; D(0) = D_0 \geq 0,
\end{cases}
\tag{2.3}
$$

where the $H(t)$, $I(t)$, $V(t)$ and $D(t)$ functions represent the number of healthy cells, the number of productively infected cells, the number of viruses released by infected cells and the dead cells at time $t$, respectively, $\beta$ denotes the healthy cells' infection rate, $\delta$ stands for the death rate of infected cells, $p$ is the virus production rate and $c$ is the virus removal rate. We emphasize the fact that unlike the hybrid PDE–ABM model, the above ODE system is not defined in space.

Technically, the $V(t)$ function can be defined to represent either the number of viruses or the virus concentration itself depending on the specific application we are considering. In the simulation results of §3, $V(t)$ stands for concentration.

Simply because of the physical meaning behind these functions, the respective initial values are naturally set in the region $\Gamma = \{(u_1, u_2, u_3, u_4) \in \mathbb{R}^4 : u_1, u_2, u_3, u_4 \geq 0\}$. For the well-posedness and boundedness of model (2.3), see appendix C.

The basic reproduction number of system (2.3) is given by

$$
\mathcal{R}_0 = \frac{p\beta H_0}{c\delta}.
\tag{2.4}
$$

For the derivation of this number and how it governs the threshold dynamics of the system, see appendix D.

## 2.3. The connection between the two main models and their parameters

In this subsection, we examine the relationship between the two main models' respective parameters. Expressing these connections is not always a trivial task as the hybrid PDE–ABM framework exists in space, while the ODE model's functions are defined as variables only in time. In order to match the two different systems' corresponding parameters, we need to fix some basic features of the spatial domain: first of all, let $A$ denote the complete area of the state space; moreover, for simplicity, let us assume that each cell has the area of a unit space—for the latter we introduce the notation $\sigma^2$. Finally, let $\tau$ denote the unit step in time.

(i) **Parameters related to cell death:** We emphasize that the two main models use different approaches to quantify the chance of an infected cell's death—on the one hand, the ODE model (2.3) works with a $\delta$ death *rate*; on the other hand, the hybrid system uses a $P_D$ *probability*. We can easily obtain a conversion between probability and rate by following the exact meaning behind these parameters. When infected cells die with a death rate $\delta$, their natural decay can be described by the function $e^{-\delta t}$; hence, the probability of an infected cell's death between any two arbitrary time points $t_1$ and $t_2$ is given by

$$
\frac{e^{-\delta t_1} - e^{-\delta t_2}}{e^{-\delta t_1}} = 1 - e^{-\delta(t_2 - t_1)}.
$$

Specifically, for a time interval of length $\tau$ the above formula means that an infected cell dies within that given time frame with a probability of $1 - e^{-\delta\tau}$.

Applying the Taylor expansion of the exponential function and combining it with the fact that $\tau$ is small, we arrive at the $1 - e^{-\delta\tau} \approx \delta\tau$ approximation, i.e. the connection between $\delta$ and $P_D$ is given by

$$
P_D \approx \delta \cdot \tau.
\tag{2.5}
$$

(ii) **Parameters related to new infections:** In this part, we establish a connection between the ODE model's infection rate $\beta$ and the hybrid system's probability of infection $P_I$. Similarly to the

previous point, we need to quantify a relationship between parameters of different dimensions—one being a *rate*, the other a *probability*—but this time the solution is a bit more complex due to the role of spatial factors. We first focus on the hybrid model's $P_I$ parameter solely within the context of the PDE–ABM system. We define the probability of a cell's infection in a way that this probability is directly proportional to the local virus concentration $V^h(\Omega_{i,j})$ in the $(i, j)$th cell (the number of viruses per unit space) and to the $\tau$ time unit, i.e. we have

$$P_I(V^h(\Omega_{i,j}), \tau) = \iota \cdot V^h(\Omega_{i,j}) \cdot \tau, \tag{2.6}$$

where $\iota$ is some appropriately set constant value. Our next step is to express the relationship between $\beta$ and $P_I$ for a specifically chosen, simplified scenario—we temporarily assume a homogeneous virus distribution over the domain $\Omega$. Now, the key to expressing $P_I$ in terms of $\beta$ consists in carefully counting the newly infected cells over one iteration in both the ODE and the hybrid systems. Assuming $H$ healthy cells and a $V$ total number of viruses at a given time, the corresponding number in the ODE model is naturally $\beta \cdot V \cdot H \cdot \tau$. When we switch to the context of the spatial hybrid model, we need to keep in mind that the virus particles are now spread throughout the entire domain $\Omega$, and as a consequence, a single cell is exposed only to the locally, physically present virus particles, whose number is $\bar{v} = V/A$. This means that the expected value of the total number of newly infected cells in the hybrid system is $H \cdot P_I(\bar{v}, \tau)$. Setting the respective values in the two main models equal leads us to

$$P_I(\bar{v}, \tau) = \beta \cdot A \cdot \bar{v} \cdot \tau. \tag{2.7}$$

The final step is to combine (2.6) and (2.7)—by substituting $V^h(\Omega_{i,j}) = \bar{v}$ in the former we immediately obtain $\iota = \beta \cdot A$. The connection between $\beta$ and $P_I$ is thus captured by

$$P_I(V^h(\Omega_{i,j}), \tau) = \beta \cdot A \cdot V^h(\Omega_{i,j}) \cdot \tau. \tag{2.8}$$

We highlight that the hybrid model's $P_I$ parameter takes the above form exclusively when the PDE–ABM model's parameters are configured with a very specific goal in mind: to match the ODE system. Otherwise, when the hybrid software is used completely as a standalone, $\iota$ is simply a parameter in the hybrid model.

(iii) **Parameters related to virus production:** The ODE model (2.3) parameter $p$ corresponds to virus production rate per unit time. Respectively, in our spatial hybrid model's virus dynamical equation (2.1) the parameter $g_{i,j}$ represents the virus production rate per unit time per unit space[2]—in particular, $g_{i,j} = f_{i,j}$ within infected cells (see (2.2)); hence, clearly, $f_{i,j}$ matches $p$.

(iv) **Parameters related to virus removal:** Analogously, the respective pair of the hybrid model's $m_V$ parameter is the ODE system's $c$ virus removal constant.

(v) **Parameters related to virus diffusion:** We highlight that the hybrid model's diffusive constant $D_V$ does not have a corresponding parameter in the ODE system, as the latter model is defined only in time and diffusivity is strictly related to spacial dimensions.

We summarize the respective parameters' relationship to each other in table 1.

## 2.4. Implementation

This section describes important technical details related to the hybrid PDE–ABM framework's implementation. Our numerical computations are based on a free and open source Java software package, HAL [10]; our source code is publicly accessible in the Github repository [12].

We note that most of our work concerning implementation is centred around customizing the generic HAL package to our concrete virological application, e.g. managing and tracking healthy, infected and dead cells, or putting certain probablistic state change approaches into practice. The deeper, application-independent segments of the HAL library on the other hand do not require any further consideration or optimization from our side; we rely on these core parts without effectuating any modifications. Just to mention one example, the numerical stability for the finite difference diffusion fields is an essential, integral part of HAL itself, and our implementation respects and preserves the original code on these levels—as a consequence, we use the implicit method that the underlying HAL library uses for dealing with the diffusion term.

We simulate the PDE–ABM model's behaviour in an $\Omega \subset \mathbb{R}^2$ two-dimensional bounded domain, hence the $\Delta V^h$ Laplacian takes the form of $V^h_{xx} + V^h_{yy}$, where $(x, y) \in \Omega$. The resolution of the state

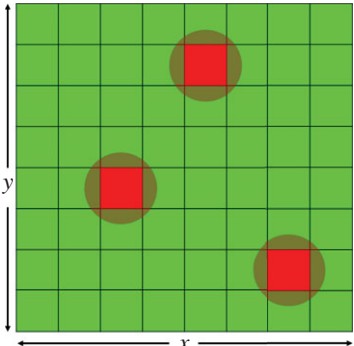

**Figure 1.** An ABM state space where most cells are healthy, while a small number of randomly infected cells start spreading the virus in the domain.

**Table 1.** Description of the hybrid PDE–ABM model's parameters and their relationship to the ODE system. Naturally, the notations [healthy cell], [infected cell] and [dead cell] represent a healthy cell, an infected cell and a dead cell—as a unit—respectively. Analogously, [virion]/$\sigma^2$ stands for the concentration unit.

| symbol | parameter | unit | value corresponding to the ODE model |
|---|---|---|---|
| $P_I$ | probability of infection for the $(i, j)$th cell | | $\beta \cdot A \cdot V^h(\Omega_{i,j}) \cdot \tau$ |
| $P_D$ | probability of death of an infected cell | | $\delta \cdot \tau$ |
| $f_{i,j}$ | virus production rate of an infected cell | $\frac{[\text{virion}]}{\sigma^2 \cdot \tau}$ | $p$ |
| $D_V$ | virus diffusion | $\sigma^2 \tau^{-1}$ | |
| $\mu_V$ | virus removal rate | $\tau^{-1}$ | $c$ |
| $H_0$ | initial number of healthy cells | [healthy cell] | $H_0$ |
| $I_0$ | initial number of infected cells | [infected cell] | $I_0$ |
| $V_0^h(x, y)$ | initial value of virus concentration | [virion]/$\sigma^2$ | $V_0$ |
| $D_0$ | initial number of dead cells | [dead cell] | $D_0$ |

$\sigma$: space unit.

$\tau$: time unit.

space is chosen to be $200 \times 200$, which means we work with a total number of 40 000 cells (this choice is mainly due to the hybrid model's computational demands—we found that concerning the number of cells, 40 000 is a reasonable value for exploring virus dynamics on an ordinary computer).

Concerning the spatial boundary, we apply Neumann boundary conditions on the edges of the hybrid state space. This approach is favourable for our investigations by guaranteeing zero flux across the boundary. The 'closed world' we get as a result is beneficial since it allows us to realistically observe the damage caused by a certain amount of infected cells without external disturbance. Wall-like edges of laboratory assays also naturally correspond to no-flux Neumann boundary conditions, and thus we expect this setting to match the outcome of *in vitro* experiments.

As for the initial conditions in our simulations, we consider the case where we have zero virus concentration in the beginning, but we do have a small number of infected cells in the domain to start with—the virus will spread from these infected cells and, as a result, more and more originally healthy cells become infected. This first generation of infected cells is chosen and distributed randomly. Each cell independently has a 0.0005 probability to be in an infected state at the beginning of the simulation, otherwise it is in a healthy state (with probability 0.9995). The number of initially infected cells is denoted by $\chi$. Figure 1 illustrates the initial state of our hybrid PDE–ABM model.

After giving a clear definition of the initial and boundary conditions in our implementation, we highlight another interesting technical detail: the internal realization of the virus production functions. At the abstract definition of the hybrid PDE–ABM model, we assumed that the virus source term in

(2.1) was continuous. This was important for certain theoretical reasons (see appendix B), but as we move on to implementation-related decisions and solutions, we make a simplifying step regarding this source function. We take a discretized approach, namely, throughout the simulations, a given cell's $g_{i,j}$ virus production function (and in particular, $f_{i,j}$) is quantified assuming that infected cells have a constant virus production rate; formally, $g_{i,j}(t, x, y) = $ a *non-zero constant* when $s_{i,j}(t) = I$ and $(x, y) \in \Omega_{i,j}$. We highlight that what really happens at the implementation process is that in each time loop and for each cell (thus, for all $(i, j) \in \mathcal{J}$) we solve equation (2.9) to obtain the virus concentration in a given cell:

$$\frac{\partial V^h(t, x, y)}{\partial t} = D_V \Delta V^h - \mu_V \cdot V^h(t, x, y) + g_{i,j}(t, x, y), \quad t > 0, \ (x, y) \in \Omega_{i,j}. \tag{2.9}$$

We also emphasize that in order to make the model as realistic as possible, we take some probabilistic considerations into account, and as a result we use a stochastic approach in the implementation of both cell infection and cell death.

— *Stochastic implementation of a healthy cell's infection:* In the $P_I = \beta \cdot A \cdot V^h(\Omega_{i,j}) \cdot \tau$ expression (obtained in (2.8)), we assume that the infection rate $\beta$ and the $A$ area of the domain are constant, but $V^h(\Omega_{i,j})$ is changing in both space and time—once the virus concentration is recalculated for a given cell, we then compare the correspondingly updated $P_I$ value with an $r_{i,j}^I$ random number, which is generated newly for each cell. In the case where the cell's $P_I$ value is greater than $r_{i,j}^I$, the cell in question gets infected.
— *Stochastic implementation of an infected cell's death:* Similarly, a random $r_{i,j}^D$ number is generated for each infected cell at each time step, and the given cell's death is determined by comparing this random number with $P_D$: naturally, the infected cell dies if $P_D$ is greater than $r_{i,j}^D$.

All of the random numbers noted above are real numbers from standard uniform distribution on the interval (0,1) and they are obtained by means of Java's random number generator.

Figure 2 shows the flow diagram of the program we implemented to simulate the virus spread and observe the spatial distributions of infected cells.

# 3. Results

We present and explain our most significant computational results in this section. In particular, we illustrate the advantages of the spatio-temporal hybrid system over the classical ODE model, and we show an example where we successfully recreate the actual results of an *in vitro* experiment assessing SARS-CoV-2 propagation. Moreover, by means of the PDE–ABM system, we explore and compare the spatial patterns of influenza and COVID-19.

## 3.1. Comparing the hybrid PDE–ABM model and the ODE model through their solutions

This section is principally dedicated to illustrating the impact of the PDE–ABM model's built-in extra information regarding spatial factors such as diffusion. In the following steps, we compare the numerical solutions of the (only time-dependent) ODE model and the hybrid spatio-temporal PDE–ABM system. We emphasize the fact that in this subsection, we consider a generic scenario that does not necessarily correspond to influenza or SARS-CoV-2 specifically. Our main goal here is to outline a comparison itself between the two main models making sure that their parameters represent the same virus infection in identical circumstances—with the very important exception of virus diffusion, which incorporates spatial effects.

As for the ODE model's simulation, we consider the following parameter values:

$$\beta = 10^{-6}, \quad \delta = 8 \times 10^{-3}, \quad p = 4 \times 10^{-2}, \quad c = 2 \times 10^{-2}. \tag{3.1}$$

Figure 3 depicts the solution of the ODE model (2.3) using the parameter values defined in (3.1) and the initial conditions $H_0 = 40\,000 - I_0$, $I_0 = 20$, $V_0 = 0$ and $D_0 = 0$.

Once the ODE parameters are set, most of the corresponding hybrid parameter values are automatically defined through the connections described in table 1. We obtain

$$P_I = 0.04 \times V^h(\Omega_{i,j}), \quad P_D = 8 \times 10^{-3}, \quad f_{i,j} = 4 \times 10^{-2}, \quad \mu_V = 2 \times 10^{-2}. \tag{3.2}$$

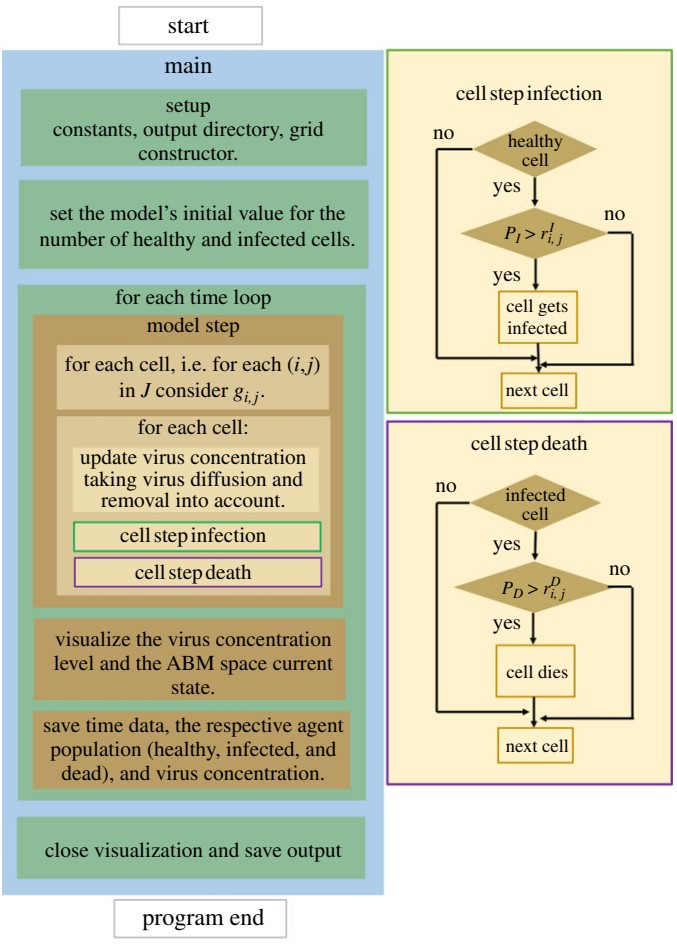

**Figure 2.** The program flow diagram of the PDE–ABM model's numerical simulation based on HAL [10].

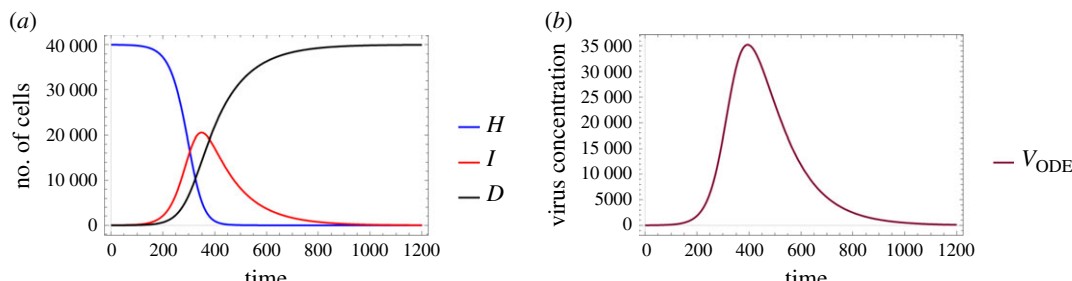

**Figure 3.** A sample solution of model (2.3) considering the parameters given in (3.1) with $H_0 = 40\,000 - I_0$, $I_0 = 20$, $V_0 = 0$ and $D_0 = 0$ as initial values. (a) Number of cells over time and (b) virus concentration over time.

Naturally—due to the fact that the ODE model is defined only in time—the hybrid system's diffusion coefficient is an exceptional parameter missing from the list above. To demonstrate the notable difference made by virus diffusion in the hybrid model's numerical solutions (and to compare the latter with the ODE model's solution), we simulate our spatial model with two different diffusion coefficients:

$$D_{V1} = 0.2 \quad \text{and} \quad D_{V2} = 20 \cdot D_{V1} = 4. \tag{3.3}$$

Figures 4 and 5 illustrate the different cell and virus dynamics emerging from scenarios that are identical apart from their diffusion coefficients. Both graphs demonstrate cell states on the left and virus spread on the right, captured in four different time points. Infected cells (denoted by red squares) are apparently distributed randomly at the initial state, and virus spread in the domain is clearly originating from infected cells. Comparing these two images we can observe the following significant differences. Firstly, as one would naturally expect, a higher diffusion coefficient results in a

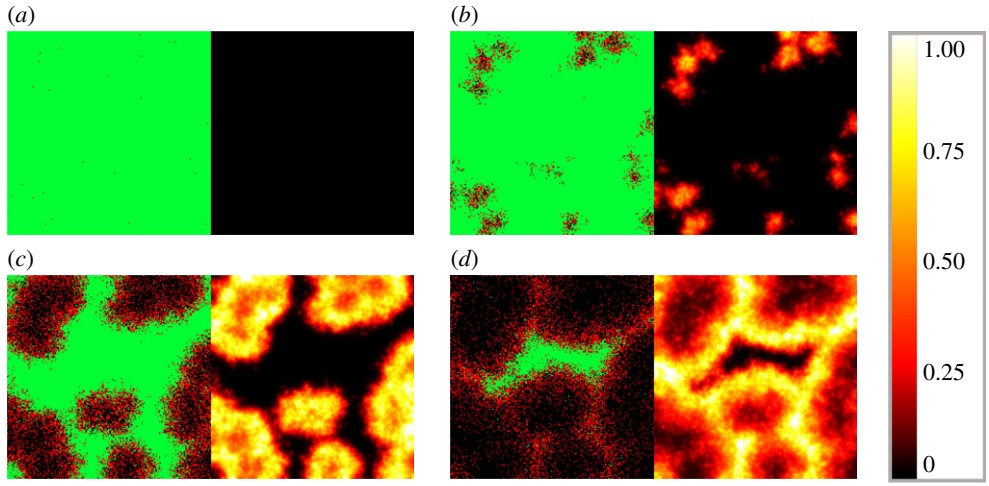

**Figure 4.** Sample hybrid PDE–ABM results considering (3.2) and $\boldsymbol{D_{V1}} = 0.2$ at $t = 0$, 240, 480 and 720 (*a–d*). Infected, healthy and dead cells (denoted by red, green and black squares, respectively) are shown on the left and virus spread is depicted on the right in all four panels (*a–d*). The colour bar is understood in virions per unit space (table 1).

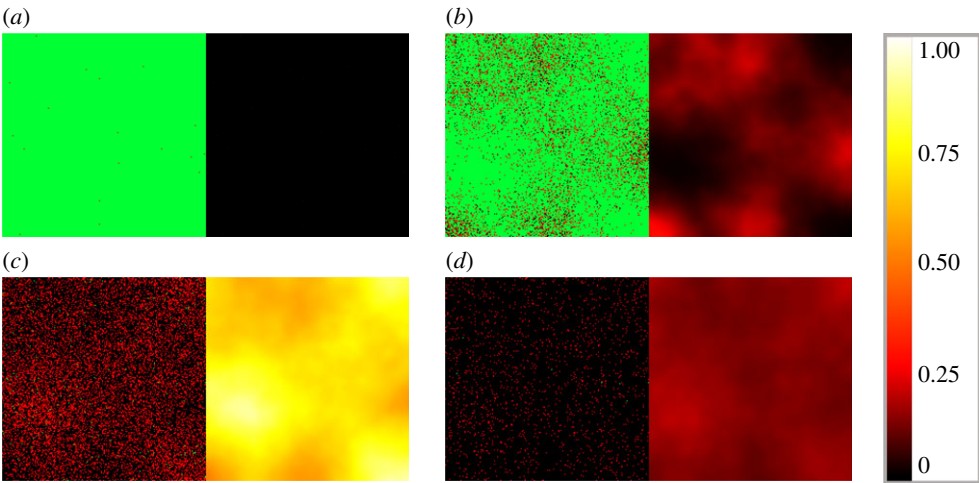

**Figure 5.** Sample hybrid PDE–ABM results considering (3.2) and $\boldsymbol{D_{V2}} = 4$ at $t = 0$, 240, 480, and 720 (*a–d*). Infected, healthy and dead cells (denoted by red, green and black squares, respectively) are shown on the left and virus spread is depicted on the right in all four panels (*a–d*). The colour bar is understood in virions per unit space (table 1).

much more homogeneous infection spread in space. In more detail, figure 4 captures a typical low-diffusion scenario where infected cells are found mainly in separate, island-resembling sets, while figure 5 shows a fundamentally different phenomenon: the layout of infected cells in the latter image is apparently much more even. An analogous difference is visible on the level of spatial virus distribution as well. On the one hand, figure 4 presents distinctive, sharp borders and well-defined lines that characterize low-diffusion virus dynamics; on the other hand, figure 5 shows—roughly speaking—foggy, blurred shades corresponding to the increased diffusion value. Our second observation is related to the total number of surviving cells by the end of the simulation: this value is clearly higher for the output obtained with a smaller diffusion coefficient, i.e. in figure 4. In other words, according to our simulations, a higher diffusion value results in a higher number of cells that get damaged by infection (i.e. cells that are either currently infected or are already dead).

Figures 6 and 7 examine identical scenarios with a different approach: they describe the corresponding virus dynamics in terms of aggregated cell numbers and virus concentration. These figures contain information in a condensed format—their clarity allows us to detect further dynamical differences caused by different diffusion values. Firstly, we can observe that the magnitude of the virus concentration peak is almost twofold bigger for a higher diffusion value (figure 7) than in the corresponding lower diffusion case (figure 6). The peak itself is reached at around $400\tau$ for $D_{V2} = 4$,

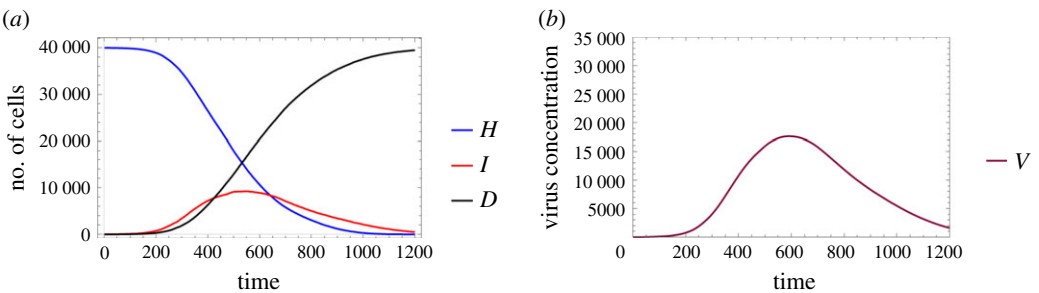

**Figure 6.** A sample solution of the hybrid PDE–ABM model considering parameters specified in (3.2) and $D_{V1} = 0.2$. (a) Number of cells over time and (b) virus concentration over time.

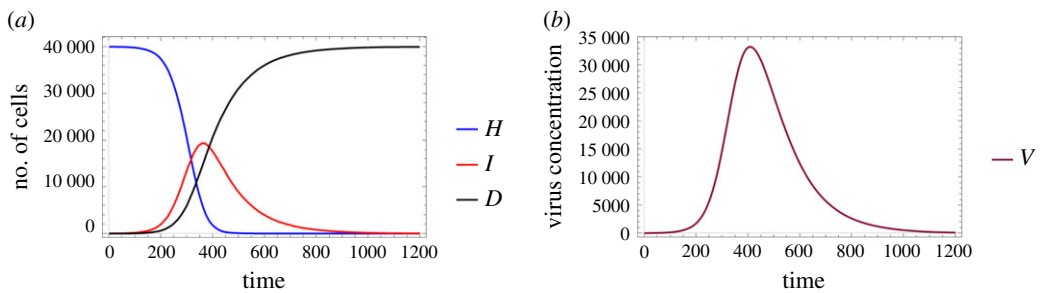

**Figure 7.** A sample solution of the hybrid PDE–ABM model considering parameters specified in (3.2) and $D_{V2} = 4$. (a) Number of cells over time and (b) virus concentration over time.

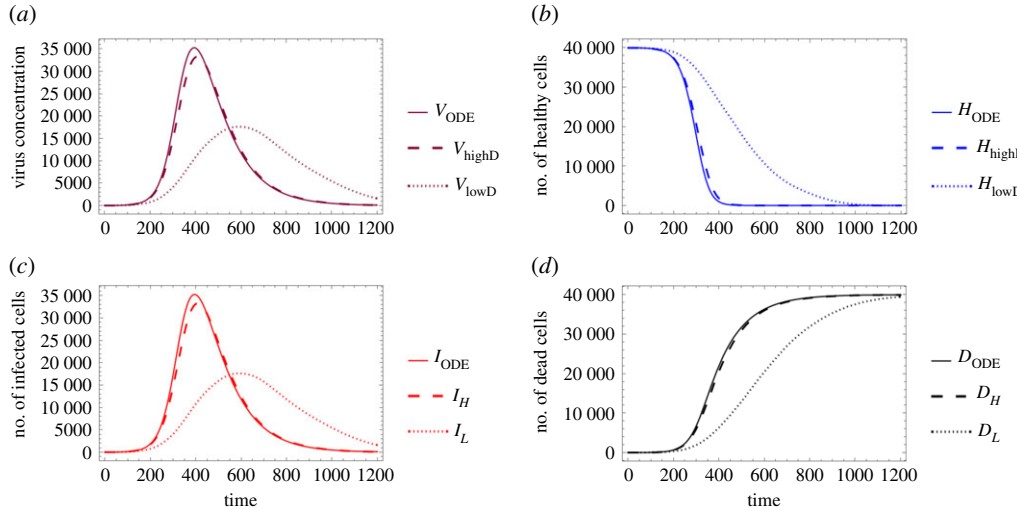

**Figure 8.** Comparison between the ODE model and the hybrid PDE–ABM system with two different diffusion values (as given in (3.3)). It is clearly seen that the outputs are very similar for a high diffusion coefficient, as opposed to the case of low diffusion coefficient. (a) Virus concentration over time, (b) number of healthy cells over time, (c) number of infected cells over time and (d) number of dead cells over time.

while for $D_{V1} = 0.2$ the analogous event takes place only after $600\tau$, i.e. in the latter case, the peak is reached approximately one and a half times slower. The sharpness of these peaks is clearly another significant difference too—a lower diffusion value seems to lead to a flatter virus concentration curve. We highlight that analogous features are visually obvious for the number of infected cells too. Finally, we observe that the faster fall in the number of healthy cells and the quicker increase in the number of dead cells in figure 7 suggest that in this particular setting a higher diffusion coefficient leads to a worse scenario overall, according to our results.

Finally, in figure 8, we compare the solution of the ODE system with the respective numerical solutions calculated for our spatial hybrid model considering the previously described two different virus diffusion values. Figure 8 illustrates a special relationship between the solutions of the two main

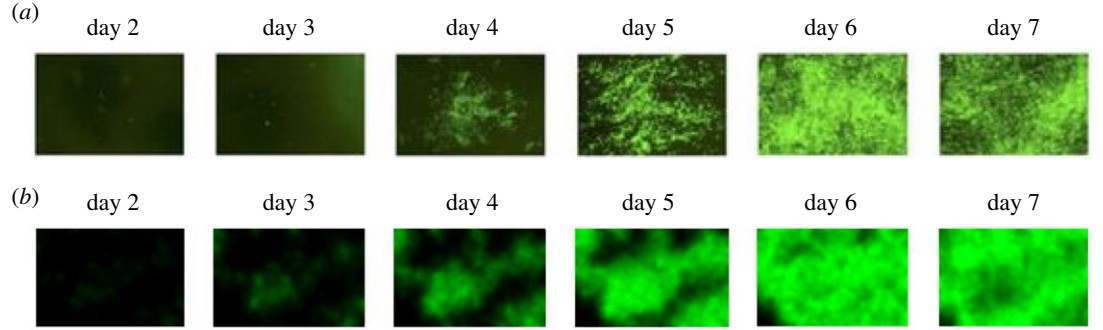

**Figure 9.** Numerical prediction of virus propagation. (*a*) Experimental *in vitro* results of [13] assessing viral spread. (*b*) The simulated spatio-temporal dynamics of SARS-CoV-2 virus spread in human airway epithelial cells—the results were obtained by our source code implementing the hybrid PDE–ABM model. This sequence of pictures from our model output shows a striking resemblance to fig. 4B in [13] (seen below), where the latter depicts real experimental results assessing viral propagation. Note the colour choice we apply in this figure: in order to match our simulation's colours to the experimental results in [13], in these particular images the colour green represents virus particles and not healthy cells.

systems: increasing the spatial model's diffusion coefficient apparently impacts the hybrid system's solution in a way that it, roughly speaking, gets closer to the ODE model's numerical solution—this corresponds to the fact that the ODE system represents a scenario where all particles can interact with any other particle, i.e. in some sense the diffusion coefficient is infinite. In other words, the lower the actual virus diffusion value is, the more important it is to consider a more complex spatio-temporal system to model the respective virus spread.

## 3.2. Applications

In one way or another, all applications of the proposed PDE–ABM hybrid model naturally focus on exploiting or predicting spatial information regarding virus spread. The first application detailed in §3.2.1 is dedicated to the latter; here we perform a simulation to obtain as much information as possible on the spatial dynamics of a COVID-19 infection and we compare our computer-generated predictions with actual results of *in vitro* experiments. On the other hand, §3.2.2 is dedicated to observing some important differences between influenza and COVID-19—these are identified more easily thanks to the additional spatial information that is integrated into the hybrid model.

### 3.2.1. Predicting the spatio-temporal spread of SARS-CoV-2 in human airway epithelial cells

Perhaps one of the most straightforward applications is the prediction of spatial virus propagation itself. In this first scenario, we simulate the spatial spread of a SARS-CoV-2 infection over the course of 7 days and we compare our results with real-life observations obtained by scientific experiments. Specifically, we consider fig. 4B in [13], where the authors examine the spread of fluorescent SARS-CoV-2 in human airway epithelial cells at the indicated days with or without peptide treatment—naturally, when we compare our results, we focus on the graphs listed in the line that corresponds to our scenario (i.e. investigating uninhibited virus propagation excluding any treatment).

Examining figure 9*b* and 9*a* (the latter is part of fig. 4B in [13]), we observe that the key events and features of virus spread match in a reassuring way: the first significant and bigger sign of infection appears at day 4, a clearly visible peak is reached at day 6, while a slight decrease in the infection's severity starts to show between day 6 and day 7. At the same time, we highlight that there is a natural limit to how accurately our simulations can recreate the specific events depicted in [13]: firstly, to our best knowledge, the exact values of virus death rate, infection rate and diffusion are not specified in [13]. Secondly, the authors have not disclosed the number of cells they worked with—this means that the resolutions of the corresponding ABM fields are different (theirs being unknown). The latter is particularly important as it is most likely one of the reasons why the computer-simulated results are less 'sharp' compared to the experimental scenario's images: roughly speaking, the generated results resemble ink spreading in water to some extent, while the *in vitro* results have an apparent grain-like structure. This matter can possibly be the subject of future investigation.

**Table 2.** Description of the parameters of ODE model (2.3). Naturally, the notations [healthy cell], [infected cell] and [dead cell] represent a healthy cell, an infected cell and a dead cell—as a unit—respectively.

| symbol | parameter | unit | value for influenza | value for COVID-19 | ref. |
|---|---|---|---|---|---|
| $\beta$ | infection rate | ml [virus copy]$^{-1}$ min$^{-1}$ | $0.94 \times 10^{-7}$ | $1.01 \times 10^{-7}$ | |
| $\delta$ | infected cell death rate | min$^{-1}$ | $2.66 \times 10^{-3}$ | $7.2 \times 10^{-4}$ | [12,14] |
| $p$ | virus production rate | [virus copy] min$^{-1}$ ml$^{-1}$ [infected cell]$^{-1}$ | $1.666 \times 10^{-2}$ | $3.72 \times 10^{-3}$ | [12,14] |
| $c$ | virus removal rate | min$^{-1}$ | $2.08 \times 10^{-3}$ | $1.67 \times 10^{-3}$ | [12,14] |
| $H_0$ | initial number of healthy cells | [healthy cell] | $40\,000 - 20$ | $40\,000 - 20$ | |
| $I_0$ | initial number of infected cells | [infected cell] | 20 | 20 | |
| $V_0$ | initial value of virus concentration | [virus copy] ml$^{-1}$ | 0 | 0 | |
| $D_0$ | initial number of dead cells | [dead cell] | 0 | 0 | |

$\tau = 1$ min.

### 3.2.2. Investigating and comparing the main properties of influenza and COVID-19

While §3.1 focused on comparing the ODE model's and the hybrid system's respective solutions in a generic scenario, the present—and in some sense, main—section's goal is to explore these solutions for two specific viruses, namely, influenza and SARS-CoV-2.

In order to investigate the propagation of an influenza or COVID-19 infection, we need to set up both of our models with parameters that represent the features of the previously mentioned viruses as accurately as possible. This is a non-trivial task not only because SARS-CoV-2 is relatively new for scientists, but also because several other small, technical, but important issues arise when we want to compare two different kinds of solutions. One particularly intriguing example is the accurate setting of the $D_V$ diffusion coefficient—the $0.65\ \mu m^2\ s^{-1} \approx 0.2\sigma^2\ min^{-1}$ value we use in the remaining part of this article is chosen from a reasonable range. We elaborate the limitations, practical considerations and further details concerning the parametrization process in appendix A. Here, we just refer to the tables containing the final values—the ODE model's parameter setting is summed up in table 2, while the parameter values we use during the simulations for the hybrid PDE–ABM model are given in table 3.

Now we are ready to compare the two different models' predictive performances. Firstly, we consider the case of influenza. Figure 10 shows the respective numerical solutions obtained by the ODE model and the hybrid system—we note that the solution given by the ODE model is similar to the one described in [16]. We also refer back to an observation we made in §3.1 regarding the diffusion coefficients: the lower this diffusion value is, the further the hybrid and ODE models' respective solutions are from each other. Finally, we highlight another important aspect in connection with figure 10. The authors of [2,17] suggest that in the upper respiratory tract about 30–50% of the epithelial cells are destroyed at the peak of infection. This corresponds to our simulated results in a reassuring way.

The second subject of our comparative investigations is COVID-19. Similarly to the concept of the previous image, figure 11 demonstrates the simulated results given by the ODE model and the PDE–ABM hybrid system, but in this case the computations were executed with the parameters representing COVID-19. We observe that the ODE model and the hybrid system generate solutions that are somewhat closer to each other than their respective solutions computed for influenza.

From this point, in this section, we examine influenza and SARS-CoV-2 propagation exclusively by means of the hybrid PDE–ABM system.

In order to compare the dynamical features of SARS-CoV-2 and influenza, we firstly simulate the viral load and the number of infected cells for both of these two viruses. The numerical solutions are

**Table 3.** Description of the hybrid PDE–ABM model's parameters. Naturally, the notations [healthy cell], [infected cell] and [dead cell] represent a healthy cell, an infected cell and a dead cell—as a unit—respectively.

| symbol | parameter | unit | value for COVID-19 | value for influenza | ref. |
|---|---|---|---|---|---|
| $P_I$ | probability of infection | | $3.76 \times 10^{-3} \times V^h(\Omega_{ij})$ | $4.04 \times 10^{-3} \times V^h(\Omega_{ij})$ | |
| $P_D$ | probability of the infected cell death | | $2.66 \times 10^{-3}$ | $7.2 \times 10^{-4}$ | |
| $f_{ij}$ | virus production from an infected cell | [virus copy] ml$^{-1}$ min$^{-1}$ [infected cell]$^{-1}$ | $1.666 \times 10^{-2}$ | $3.72 \times 10^{-3}$ | |
| $\mu_V$ | virus removal rate | min$^{-1}$ | $2.08 \times 10^{-3}$ | $1.67 \times 10^{-3}$ | |
| $D_V$ | virus diffusion | $\mu m^2 \ s^{-1}$ | $[3.18 \times 10^{-3}, 3.18]$ | $[3 \times 10^{-6}, 3]$ | [11,15] |
| | | $\sigma^2$ min$^{-1}$ | $[0.97 \times 10^{-3}, 0.97]$ | $[0.9 \times 10^{-6}, 0.9]$ | appendix A |
| $H_0$ | initial number of healthy cells | [healthy cell] | $40\,000 - \chi$ | $40\,000 - \chi$ | |
| $I_0$ | initial number of infected cells | [infected cell] | $\chi$ | $\chi$ | |
| $V_0^h(x,y)$ | initial value of virus concentration | [virus copy] ml$^{-1}$ cell$^{-1}$ | $0$ | $0$ | |
| $D_0$ | initial number of dead cells | [dead cell] | $0$ | $0$ | |

$\sigma = 14 \ \mu m$.

$\tau = 1$ min.

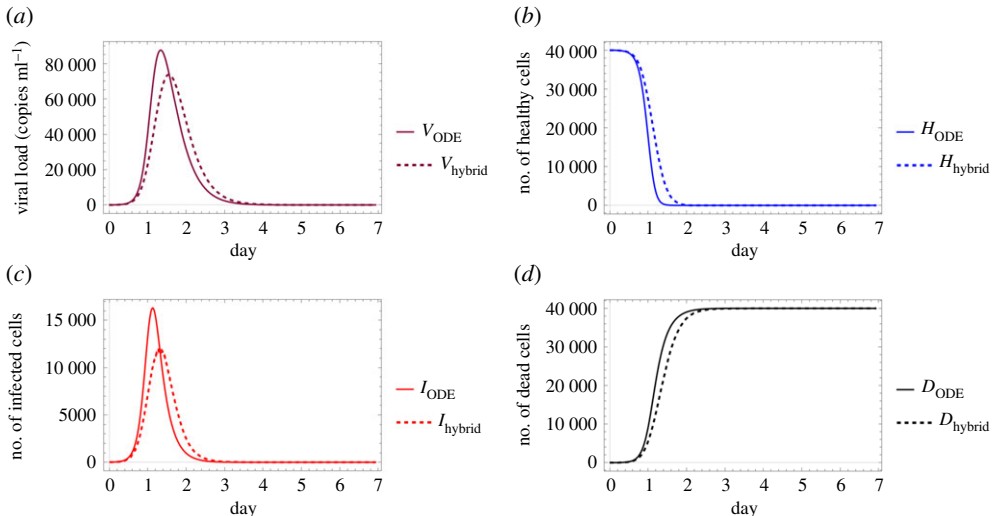

**Figure 10.** Comparison of the ODE model (solid lines) and the hybrid PDE–ABM model (dashed lines) for influenza using the parameters in tables 2 and 3, applying $D_V = 0.2\sigma^2$ min$^{-1}$ in the hybrid model. All panels depict a change taking place over the course of 7 days; specifically, (a) viral load, (b) number of healthy cells, (c) number of infected cells and (d) number of dead cells.

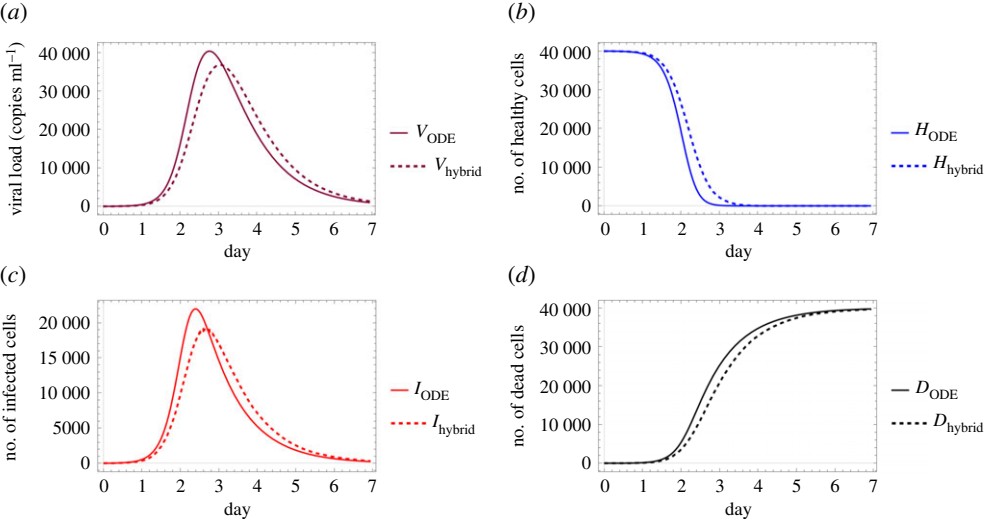

**Figure 11.** Comparison of the ODE model (solid lines) and the hybrid PDE–ABM model (dashed lines) for SARS-CoV-2 using the parameters in tables 2 and 3, applying $D_V = 0.2\sigma^2$ min$^{-1}$ in the hybrid model. All panels depict a change taking place over the course of seven days; specifically, (a) viral load, (b) number of healthy cells, (c) number of infected cells and (d) number of dead cells.

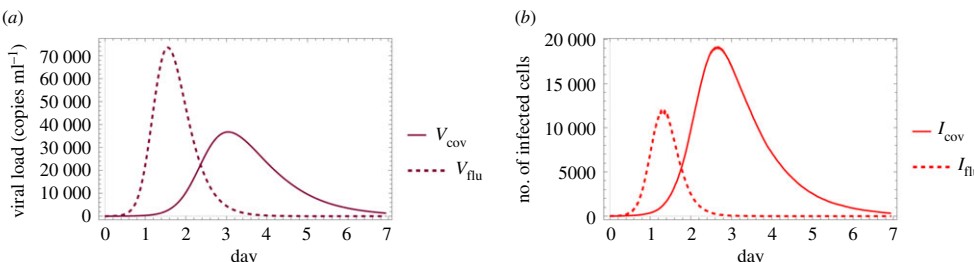

**Figure 12.** The difference between influenza and SARS-CoV-2 infections. The viral load (a) and the number of infected cells (b) simulated for influenza (dashed line) and COVID-19 (solid line) using the hybrid PDE–ABM model with $D_V = 0.2\sigma^2$ min$^{-1}$.

presented in figure 12 (the respective functions for the two different viruses are shown together). First of all, we highlight that while the solutions depicted in this image are only time-dependent functions, the advantage of the hybrid PDE–ABM model is present even behind this particular result—the fact that these numerical solutions were calculated by means of the spatio-temporal hybrid model ensures that

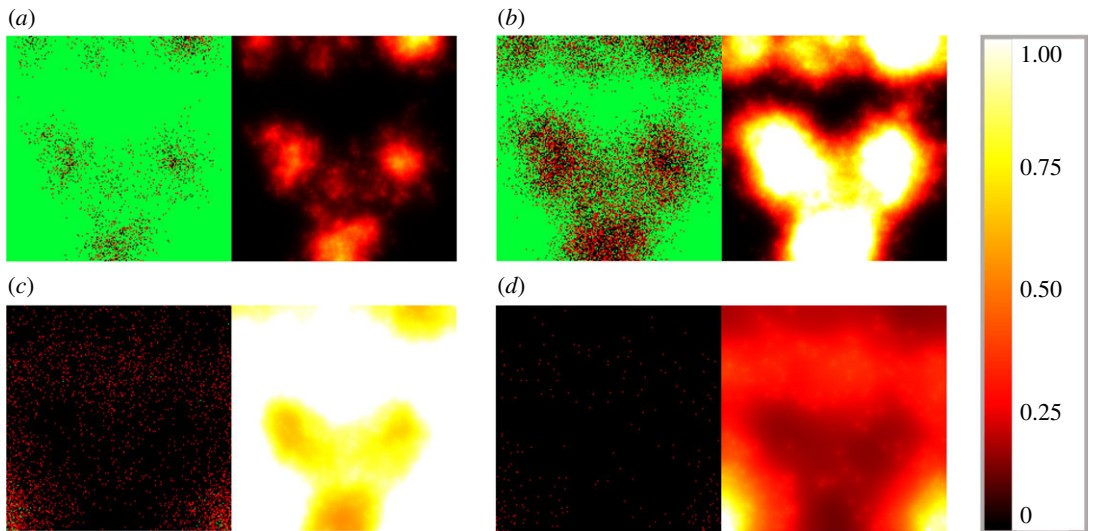

**Figure 13.** Simulated spatio-temporal numerical solutions captured (*a*) 16 h, (*b*) 24 h, (*c*) 48 h and (*d*) 64 h after influenza infection. The computations were performed by means of the hybrid PDE–ABM model using $D_V = 0.2\sigma^2 \min^{-1}$. Infected, healthy and dead cells (denoted by red, green and black squares, respectively) are shown on the left, while virus spread is depicted on the right in all four panels (*a*–*d*). The colour bar is understood in virus copies per ml per cell (table 3).

their respective values take important physical aspects such as diffusion into account. In particular, we refer to [18] for connecting some of our results to real data and describing the course of influenza infection: this work reports that virus shedding increases sharply between 0.5 and 1 day after challenge and peaks at day 2, while the average duration of viral shedding is 4.8 days. Our computer simulated results shown in figure 12 for influenza clearly match with the experimental data of [18].

Some of the results in figure 12 might seem surprising at first sight. Figure 12*a* demonstrates that in terms of viral load the peak occurs sooner for influenza, and the value itself taken at this peak is much higher too for influenza compared to COVID-19. On the other hand, if we consider the number of infected cells, figure 12*b* represents temporal behaviours—and in particular peak values—that in some sense seem like the opposite of the previous observations made for virus concentration. In more detail, the peak of the number of infected cells is much higher for COVID-19 than the respective value for influenza; however, in terms of time to peak, influenza remains the 'faster' virus out of the two by reaching its maximum in less than 2 days.

The intriguing features pointed out above will be addressed in more detail after the following discussion of spatial patterns.

Figures 13 and 14 demonstrate the distributional patterns of influenza and SARS-CoV-2 propagation. Comparing figure 13 with figure 14, it is apparent that while both COVID-19 and influenza propagation tend to show spatial patterns that, roughly speaking, resemble flocks or well-outlined explosions, the sharpness itself of this phenomenon is slightly weaker in the case of COVID-19. In particular, we refer to figure 13*b* versus figure 14*c*: if these images are approached as terrain levels, then the land described by influenza generally seems to have larger gradients in terms of topographic contour lines. We note that this becomes even more apparent for higher diffusion values, where we see rather homogeneous, blurry spatial patterns for SARS-CoV-2, while influenza infection preserves sharper frontlines. We provide supplementary images dedicated to exploring the model's sensitivity to different diffusion parameters in our public github repository [12], where we also share further visual data to grasp the infection dynamics.

We finish this section with emphasizing the significance of the hybrid PDE–ABM model by revisiting the results presented in connection with figure 12 and observing them in a new light—in particular, we explore how the detailed spatio-temporal layouts given in figures 13 and 14 contribute to a better understanding of the original functions of (and questions behind) figure 12.

As we have seen in the discussion of figure 12, the number of infected cells have a significantly smaller peak value for influenza compared to COVID-19. The new information regarding spatial patterns in figures 13 and 14 gives insight into what really happens in the background. The fact that the ratio of red and black agents can be so high for SARS-CoV-2, while it always remains relatively low for influenza suggests that influenza-infected cells do not live nearly as long as cells infected by SARS-CoV-2 (in fact, this is supported by the $P_D$ values of table 3).

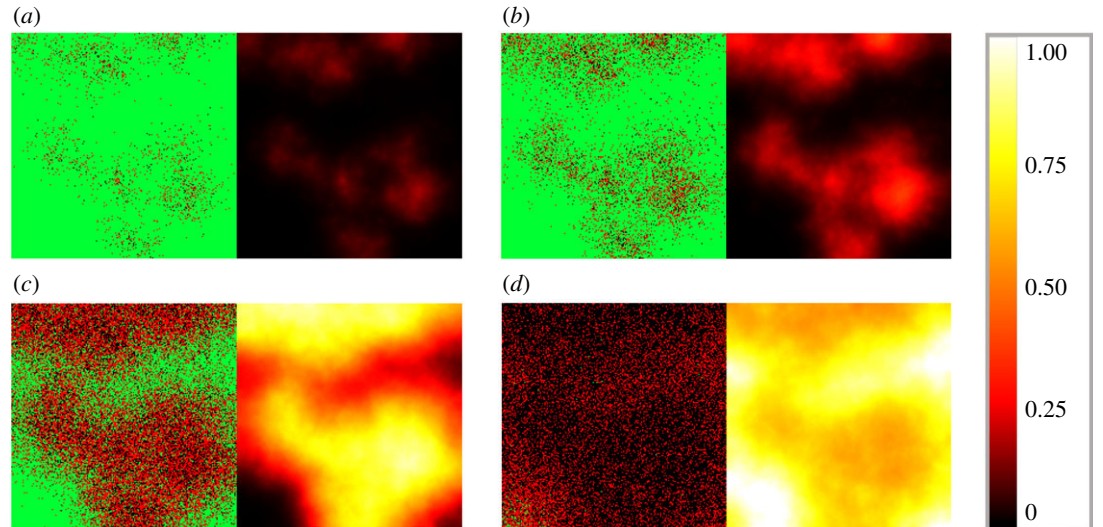

**Figure 14.** Simulated spatio-temporal numerical solutions captured (*a*) 32 h, (*b*) 40 h, (*c*) 56 h and (*d*) 88 h after SARS-CoV-2 infection. The computations were performed by means of the hybrid PDE–ABM model using $D_V = 0.2\sigma^2$ min$^{-1}$. Infected, healthy and dead cells (denoted by red, green and black squares, respectively) are shown on the left, while virus spread is depicted on the right in all four panels (*a*–*d*). The colour bar is understood in virus copies per ml per cell (table 3).

Carefully comparing figures 13 and 14 can also help us identify a reason why the (quick) viral load peak is higher for influenza in figure 12 when its cell-analogue is clearly higher for COVID-19. We begin by exploring what happens on the level of virus concentration when we have (approximately) the same amount of infected cells. In particular, figures 13*b* and 14*b* are both scenarios where we seem to have approximately the same amount of infected cells, and at the same time the difference between the corresponding virus concentration images is strikingly apparent. This suggests that the key feature behind the originally mentioned phenomenon might be a difference between the strength of the respective viral source functions—this is in fact verified by the $f_{i,j}$ values of table 3. Hence, these observations are significant not just in themselves, but also examples of biological insights provided by the hybrid PDE–ABM model.

## 4. Discussion

In this paper, we have applied two different models for investigating the dynamical aspects of virus spread. The first model we considered was a hybrid PDE–ABM system, which is essentially a result of merging a discrete state space representing epithelial cells with a continuous reaction–diffusion equation grasping virus concentration. At the same time, we have used the so-called May–Nowak system—a well-known version of the classical ODE model—as a reference system. As for theoretical completeness, we provide a rigorous analysis of both models in the appendices, including a well-posedness result related to the hybrid model and the study of the ODE model's temporal dynamics.

The hybrid model's computational implementation and the careful exploration of its results are in some sense the heart of our work—we highlight that our program code is based on a free and open source Java library, HAL [10], commonly used for oncology modelling.

Compared to the ODE system, both the decisive advantages and the main difficulties of the hybrid model are naturally related to the PDE–ABM system's added (and quite high-level) complexity: the inclusion of spatial effects. On the one hand, the limiting factors of this model include an increased computational demand and the fact that it is virtually impossible to consider a really large number of cells on an ordinary computer (we worked with a slice of tissue consisting of $4 \times 10^4$ cells). On the other hand, however, this hybrid model provides us with the invaluable spatial distribution of infection spread: by running simulations for influenza and SARS-CoV-2 propagation, the results of our spatio-temporal PDE–ABM system suggested that influenza seems to generate sharper frontlines in virus concentration than COVID-19 does; moreover, especially for higher diffusion values, COVID-19 visibly spreads in a more homogeneous manner compared to influenza. This simply would not have been possible using the ODE model as the latter is defined only in time. The ODE system

represents a scenario where all cells can interact with all virus particles, or in other words, it implicitly assumes an infinite diffusion coefficient in some sense. Real-life viruses however clearly have a finite diffusion rate. This also means that if the specific virus in question has a relatively low diffusion rate, then the ODE model's predictions regarding infection dynamics will be less accurate: the lower the diffusion coefficient, the more important it is to apply the more complex and more suitable spatial hybrid model. This phenomenon can be observed in figure 8: a small diffusion value results in great differences between the respective solutions of the PDE–ABM system and the ODE model (2.3), while the solutions are indeed close to each other for a larger diffusion value.

In terms of verifying the accuracy and correctness of our proposed model, we highlight the results of §3.2.1—we have relatively successfully recreated the real results of a scientific *in vitro* experiment: our computer-simulated results matched the actual events and features of infection spread on a satisfactory level. Regarding correctness, we refer to figure 10 as well: about 30% to 50% of the epithelial cells are destroyed in the upper respiratory system at the peak of infection, which corresponds to the observations of [2,17].

As for possible further improvements and applications of our hybrid spatial model, we mention two main points. Firstly, fine-tuning features such as immune response processes, time delay between infection and virus production, and the phenomenon of cell regeneration are ignored in our current study. These can be the subject of possible future work, although we note that the present model itself can also be considered to be highly realistic in specific cases where some of the above mentioned elements are naturally negligible (e.g. at the short early phase of an infection the immune system has typically not responded yet, while the time frame is too short for cell regeneration to be relevant). Secondly, we plan to apply the hybrid system for parameter fitting analogously as [2] used the ODE model for a similar task for the case of influenza A. In more detail, the authors of [2] calculated a best fit of the ODE model using experimental data on viral load—they extracted viral kinetic parameters such as infection rate, virus production rate, viral clearance rate and the half-life of free infectious viruses. We simulated the corresponding scenario with both systems and—as expected, considering the relatively low diffusion value set for influenza—there was an apparent difference between the respective numerical solutions of the ODE model and the hybrid system. According to our results, [2] somewhat underestimates the $R_0$ value: as figure 10 shows, in order to obtain a solution with the PDE–ABM model that corresponds to the ODE solution (and hence to the real curve), it seems that the $R_0$ value of the hybrid model needs to be higher than the value estimated by [2] to fit the experimental data. This is another example of how the assumption of homogeneous virus spread can be misleading—the kinetic values obtained by Baccam *et al.* [2] could be adjusted towards their real biological value by means of the hybrid PDE–ABM model. Thus, parameter estimation and fitting the stochastic hybrid model to various virological data is something we also consider as valuable future work.

The complex hybrid approach allows our model to capture fundamental physical processes such as diffusion. We have seen that this is paramount in analysing spatio-temporal virus spread, but we emphasize that virus diffusion itself is not the only example for this feature's significance. Future works using this framework may consider immune response or antiviral drugs. For the latter, drug diffusion is essential, since spatial heterogeneity naturally arises as the drug enters the tissue through the capillary network. Hence, the diffusive property has a key role in the analysis of antiviral drug effectiveness, which can be precisely evaluated only in a spatio-temporal context, and our proposed model can be of great use for assessing potential COVID-19 treatment strategies.

Our final synopsis is that the hybrid PDE–ABM model is better suited for thorough and detailed virus spread assessment than the classical ODE system. Following virus propagation on an individual cellular level and taking important spatial effects into account result in a more accurate and complete picture regarding the infection's outcome. Even though the additional integrated details clearly come at a price in terms of computational demand, this pays off very well in the form of information on spatial virus distribution and more accurate predictions.

Data accessibility. Data and relevant code for this research work are stored in GitHub: https://github.com/ sadeghmarzban/Hybrid-PDE-ABM-for-viral-dynamics and have been archived within the Zenodo repository: https://doi.org/10.5281/zenodo.4758518.

Authors' contributions. Conceptualization: G.R.; theoretical analysis: R.H., G.R.; software development, numerical simulations, visualization: S.M.; supervision: G.R.; literature review: all authors; writing: N.J.; mathematical modelling: all authors; reviewing and editing: all authors.

Competing interests. We declare we have no competing interests.
Funding. R.H. acknowledges the financial support from Youth Foundation of Zhejiang University of Science and Technology (grant nos. 2021QN001, 2021QN046). S.M. and R.H. were supported by Hungarian grant NKFIH KKP

129877. N.J. was supported by EFOP-3.6.1-16-2016-00008 and TUDFO/47138-1/2019-ITM. G.R. was supported by NKFIH FK 124016 and 2020-2.1.1-ED-2020-0000.

# Appendix A. Technical details related to parameter values

This appendix is dedicated to important practical details related to the calculation of the numerical data presented in tables 2 and 3.

We begin with the ODE model's parameter setting which is summed up in table 2. The parameters representing influenza and COVID-19 are defined according to [2,14], respectively. We detail two technical considerations we used during the parametrization process. (i) Firstly, in order to make sure that the respective results of the ODE system and the hybrid model are comparable, we need to initialize them with the same number of healthy cells. Due to the high computational load of the latter system, this number is naturally limited—in our case, we use $H_0 = 4 \times 10^4$. However, this number is lower than the number of healthy cells used in [2,14], which is problematic as it affects the value of $R_0$. Hence, to counterbalance the impact of the different $H_0$ value, we modify the value of $\beta$ correspondingly. (ii) Secondly, the viral load for influenza in [2] is given in $(\text{TCID}_{50}\ \text{ml}^{-1})$ units, while [14] specifies the viral load index for COVID-19 in copies $\text{ml}^{-1}$. To overcome this, we apply the results of [19], we assume that a $\text{TCID}_{50}$ unit corresponds to $2 \times 10^3$ viral copies and we change the $\beta$ and $p$ values in the influenza parameter setting correspondingly.

As for the hybrid PDE–ABM model, the exact parameter values we use during the simulations are given in table 3. With the exception of the diffusion coefficient, all of the hybrid model's parameter values have been calculated indirectly: we used the ODE parameters as a starting point and we combined these with the results detailed in table 1—knowing the formal connection between the respective ODE and PDE–ABM parameters we can easily obtain the hybrid parameters in question.

We also point out a significant difference between the units of table 1 (§2) and table 3 (§3.2). While in the more theoretical §2, we conveniently used abstract units such as $\sigma^2/\tau$, in §3.2.2 dedicated to specific applications all units become concrete (e.g. $\mu\text{m}^2\ \text{min}^{-1}$). This raises an interesting technical question—the relationship between the real-life space unit $\mu$m and the abstract space unit $\sigma$ still internally used within the implementation. In the present article, we use the physiologically accurate $\sigma = 14\ \mu$m setting.

Finally, we highlight two intriguing aspects of the diffusion coefficients.

(i) **Uncertainty.** There is scarce information on the diffusion coefficients in the literature, and the specific values vary depending on environmental factors as well. As a guideline, in table 3, we use the estimates of [11,15]. Due to the similar size of virions, we apply the same $D_V$ value—specifically, $0.65\ \mu\text{m}^2\ \text{s}^{-1}$—in our simulations both for influenza and SARS-CoV-2. The sensitivity to the respective parameter choice is explored through the supplementary images available in our github repository [12].

(ii) **Unit conversion.** Diffusion values are normally given in units $\mu\text{m}^2\ \text{s}^{-1}$. To be used in HAL [10], the $D_V$ values need to be converted into dimensionless numbers, adjusting them to the respective units used by the software. As the unit space $\sigma$ in HAL corresponds to $14\ \mu$m, we have

$$D_V\ \mu\text{m}^2\ \text{s}^{-1} = \frac{60}{14^2} D_V \sigma^2\ \text{min}^{-1}. \tag{A 1}$$

In our simulations (see [12]), we investigated the dynamics generated by diffusion values of $0.1, 0.2, \ldots, 0.9\sigma^2\ \text{min}^{-1}$—in particular, the results communicated in the main text were obtained with $0.2\sigma^2\ \text{min}^{-1} \approx 0.65\ \mu\text{m}^2\ \text{s}^{-1}$.

# Appendix B. The well-posedness and ultimate boundedness of the PDE model

In this appendix, we establish the global existence and ultimate boundedness of the virus concentration function described by equation (2.1). We make the following suitable assumption:

$$(\mathbf{H_1}): g_{i,j}(t, x, y) \in C^{(\alpha/2),\alpha}([0, \infty) \times \bar{\Omega}) \cap L^\infty((0, \infty) \times \Omega)$$

is non-negative for each $(i, j) \in \mathcal{J}$ and $\alpha \in (0, 1)$.

**Theorem B.1 (Well-posedness and ultimate boundedness).** *Let $\Omega \subset \mathbb{R}^2$ be a bounded domain with smooth boundary. Suppose that the parameters $D_V$, $\mu_V$ are positive. Then for a non-negative initial value function $V_0^h(x, y) \in C^0(\bar{\Omega})$ system (2.1) has a unique non-negative global solution $V^h(t, x, y)$ defined on $[0, \infty) \times \bar{\Omega}$. Moreover, the solution $V^h(t, x, y)$ is ultimately bounded and satisfies* $\limsup_{t \to \infty} \max_{\bar{\Omega}} V^h(t, x, y) \le (B/\mu_V)$, *where B is defined in (B 1).*

*Proof.* We use the lower–upper solution method for our proof. According to **(H₁)**, we define

$$\| \sum_{(i,j) \in \mathcal{J}} g_{i,j}(t, x, y)\|_{L^\infty((0,\infty) \times \Omega)} =: B. \tag{B 1}$$

Let $\bar{V}(t, x, y) = (\sup_{\bar{\Omega}} V_0^h(x, y) - (B/\mu_V))e^{-\mu_V t} + \dfrac{B}{\mu_V} =: V^\star(t)$ and $\underline{V}(t, x, y) = 0$. Since

$$\frac{\partial \bar{V}(t, x, y)}{\partial t} - D_V \Delta \bar{V}(t, x, y) + \mu_V \bar{V}(t, x, y) - \sum_{(i,j) \in \mathcal{J}} g_{i,j}(t, x, y)$$

$$\ge \frac{\mathrm{d}V^\star(t)}{\mathrm{d}t} + \mu_V V^\star(t) - B = 0 \tag{B 2}$$

and

$$\frac{\partial \underline{V}(t, x, y)}{\partial t} - D_V \Delta \underline{V}(t, x, y) + \mu_V \underline{V}(t, x, y) - \sum_{(i,j) \in \mathcal{J}} g_{i,j}(t, x, y)$$

$$= - \sum_{(i,j) \in \mathcal{J}} g_{i,j}(t, x, y) \le 0. \tag{B 3}$$

Clearly, the boundary condition satisfies $\partial \bar{V}/\partial \nu = 0 \ge 0 = \partial \underline{V}/\partial \nu$ and the initial condition satisfies $\bar{V}(0, x, y) = \sup_{\bar{\Omega}} V_0^h(x, y) \ge 0 = \underline{V}(0, x, y)$. Thus $(\bar{V}(t, x, y), \underline{V}(t, x, y))$ is a pair of coupled upper solution and lower solution of system (2.1). From **(H₁)**, we get that the function $f(t, x, y, V^h) = -\mu_V V^h + \sum_{(i,j) \in \mathcal{J}} g_{i,j}(t, x, y)$ is Hölder continuous with exponent $\alpha$ with respect to $t$ and $x$, $y$ on $\bar{Q}_T \times [m, M]$, and $f_{V^h}(t, x, y, V^h) = -\mu_V \in C(\bar{Q}_T \times [m, M])$, where $Q_T = (0, T] \times \Omega$, $m = \min_{\bar{Q}_T} \underline{V}(t, x, y)$ and $M = \max_{\bar{Q}_T} \bar{V}(t, x, y)$. In view of [20, theorem 2.4.6], system (2.1) has a unique solution defined on $(0, T] \times \Omega$ satisfying

$$0 \le V^h(t, x, y) \le V^\star(t). \tag{B 4}$$

By the arbitrariness of $T$, the solution of system (2.1) exists globally in time. Clearly, from the definition of $V^\star(t)$, we know that $V^\star(t) \le \max\{(B/\mu_V), \sup_{\bar{\Omega}} V_0^h(x, y)\}$. Then it follows from (B 4) that $\limsup_{t \to \infty} \max_{\bar{\Omega}} V^h(t, x, y) \le (B/\mu_V)$, which ends the proof of the theorem. ∎

In fact, since $\sum_{(i,j) \in \mathcal{J}} g_{i,j}(t, x, y) \in C^{(\alpha/2), \alpha}([0, \infty) \times \bar{\Omega})$, it follows from the regularity theory of parabolic equations that system (2.1) has a unique classical solution $V^h(t, x, y) \in C^{1+(\alpha/2), 2+\alpha}([0, \infty) \times \bar{\Omega})$. Here, the lower–upper solution method is mainly employed to estimate the upper bound of virus particles in the whole spatial domain $\Omega$ as time evolves.

# Appendix C. The well-posedness and boundedness of the ODE model

**Theorem C.1.** *Let us suppose $(H_0, I_0, V_0, D_0) \in \Gamma$ for our initial data. Then system (2.3) has a unique solution $(H, I, V, D) \in \Gamma$, i.e. $\Gamma$ is positively invariant. Furthermore, the solution of system (2.3) satisfies*

$$\left.\begin{aligned} &0 \le H(t) \le H_0, \quad 0 \le \limsup_{t \to \infty} V(t) \le \frac{p(N - V_0)}{c}, \\ &0 \le \limsup_{t \to \infty} I(t) \le (N - V_0)\min\{1, (p\beta H_0/c\delta)\}, \quad 0 \le D(t) \le N - V_0, \end{aligned}\right\} \tag{C 1}$$

*where $N = H_0 + I_0 + V_0 + D_0$.*

*Proof.* **Existence and uniqueness.** By the continuity of the right-hand side of equation (2.3), system (2.3) has at least one solution in $[0, b)$ with $0 < b \le \infty$. Since the term on the right-hand side of system (2.3) satisfies the local Lipschitz property with respect to $H, I, V, D$ in $\Gamma$, the uniqueness of solutions follows from the standard theory of ordinary differential equations.

**Positive invariance and boundedness.** For $H \ge 0$, $I \ge 0$, $V \ge 0$, $D \ge 0$, since

$$\dot{H}(t)|_{H=0} = 0 \ge 0, \quad \dot{I}(t)|_{I=0} = \beta HV \ge 0, \quad \dot{V}(t)|_{V=0} = pI \ge 0, \quad \dot{D}(t)|_{D=0} = \delta I \ge 0, \tag{C 2}$$

it follows from [21, theorem 5.2.1] that the solution $(H(t), I(t), V(t), D(t))$ of system (2.3) is non-negative for all $t \geq 0$ whenever $H_0 \geq 0$, $I_0 \geq 0$, $V_0 \geq 0$, $D_0 \geq 0$. Therefore, $\Gamma$ is positively invariant with respect to system (2.3).

Since $\dot{H}(t) = -\beta H(t) V(t) \leq 0$, we get $0 \leq H(t) \leq H_0$. Also from system (2.3), we obtain

$$\frac{\mathrm{d}(H(t) + I(t) + D(t))}{\mathrm{d}t} = 0 \quad \Rightarrow H(t) + I(t) + D(t) = N - V_0. \tag{C 3}$$

As $H(t) \geq 0$, $D(t) \geq 0$, we get $I(t) \leq N - V_0$. From the third equation of system (2.3), we have

$$\frac{\mathrm{d}V(t)}{\mathrm{d}t} \leq p(N - V_0) - cV,$$

which implies that

$$\limsup_{t \to \infty} V(t) \leq \frac{p(N - V_0)}{c}. \tag{C 4}$$

From (C 4), we know that for any $\varepsilon > 0$ there exists $T(\varepsilon) > 0$, such that when $t \geq T(\varepsilon)$, $V(t) \leq (p(N - V_0)/c) + \varepsilon$. Thus

$$\frac{\mathrm{d}I(t)}{\mathrm{d}t} \leq \beta H_0 \left( \frac{p(N - V_0)}{c} + \varepsilon \right) - \delta I.$$

By the comparison principle of ordinary differential equations, we get

$$\limsup_{t \to \infty} I(t) \leq \frac{p\beta H_0(N - V_0)}{c\delta}. \tag{C 5}$$

Since $I(t) \leq N - V_0$, we get $\limsup_{t \to \infty} I(t) \leq (N - V_0)\min\{1, (p\beta H_0/c\delta)\}$. This completes the proof. ∎

# Appendix D. The threshold dynamics of the ODE model

The basic reproduction number $\mathcal{R}_0$ can be deduced by simple reasoning: a single infected cell produces virus with rate $p$ during its expected lifetime $1/\delta$, summing up to $p/\delta$ viruses. Virus particles generate infected cells (in a healthy cell population) with rate $\beta H_0$, in an expected time period $1/c$. Overall, we find the number of new infected cells originated from the initial cell as given in (2.4). For a more formal derivation, one can also use the next generation approach [22].

**Theorem D.1.** If $\mathcal{R}_0 > 1$, then the disease-free equilibrium of the form $e_* = (H_0, 0, 0, D_s)$ is unstable, where $H_0 > 0$, $D_s$ being an arbitrary non-negative constant. This means an infection can be established. Furthermore, system (2.3) cannot undergo a Hopf bifurcation around $e_*$.

*Proof.* The Jacobian matrix $J_*$ around the disease-free equilibrium $e_*$ is given by

$$J_* = \begin{bmatrix} 0 & 0 & -\beta H_0 & 0 \\ 0 & -\delta & \beta H_0 & 0 \\ 0 & p & -c & 0 \\ 0 & \delta & 0 & 0 \end{bmatrix}. \tag{D 1}$$

Thus the characteristic equation can be calculated as

$$\lambda^2[\lambda^2 + (c + \delta)\lambda + c\delta - p\beta H_0] = 0, \tag{D 2}$$

which has eigenvalues $\lambda_{1,2} = 0$, $\lambda_{3,4} = (-(c + \delta) \pm \sqrt{(c + \delta)^2 - 4(c\delta - p\beta H_0)})/2$. Clearly, if $\mathcal{R}_0 > 1$ then $\lambda_3 > 0$ and $\lambda_4 < 0$, i.e. the characteristic equation (D 2) has a positive real eigenvalue, which indicates that $e_*$ is unstable. In addition, it is easy to see that there are no pairs of purely imaginary roots in characteristic equation (D 2) for any variational parameter, which implies that a Hopf bifurcation cannot occur. This completes the proof of the theorem. ∎

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
