## [Peer Review File · Royal Society Open Science]

Review History

RSOS-210787.R0 (Original submission)

Review form: Reviewer 1

Is the manuscript scientifically sound in its present form?

Yes

Are the interpretations and conclusions justified by the results?

Yes

Is the language acceptable?

Yes

Do you have any ethical concerns with this paper?

No

Have you any concerns about statistical analyses in this paper?

No

Recommendation?

Accept with minor revision (please list in comments)

Comments to the Author(s)**SUMMARY**

These authors create a hybrid agent based model to simulate influenza and SARS-CoV-2 and compare outputs to an ODE model. They claim that investigating the spatiotemporal dynamics provides new insights that are not captured with the ODE model. This manuscript was easy to read and I thought the approach was interesting. However, I thought there needed to be a little more discussion on why the results from the hybrid model are important and/or potentially useful for treatment (discussed below).

COMMENTS ON INTRODUCTION

- Page 4 Line 29-33: You claim that the “most urging current biological questions are already well-known.” Could you please clarify what these questions are, or what you mean by this?

COMMENTS ON RESULTS/DISCUSSION

- Page 12 Line 43-46: Define a specific time point at which the higher diffusion value results in a higher number of infected cells, since it isn't true for all time points.
- Page 12 Line 47-49: This paragraph needs more description related to what is different between Figures 6 and 7 (for example, timing and sharpness of peaks, magnitude, etc.).
- How much variability was there between simulations? For example, were there ever any cases where healthy cells did not die out because the infection went extinct?
- A paragraph or two is needed to describe why these results are important. In the abstract, the authors claim that “we can gain insight into why the outcomes of these two infections are different.” However, the authors should describe more about what is known to be different between the two infections. For example, how do their R_0 's compare? How does the spatial model provide insight into the differences seen in Figure 12? Furthermore, the authors claim that “spatiotemporal information ... is vital from the aspect of disease treatment” but do not elaborate on how these results can help motivate potential treatment strategies. Please describe a bit more on this.

COMMENTS ON METHODS

- Page 8 Line 43-33: Describe why you use no-flux boundary conditions for the ABM. Do you expect results to be dependent on choice of boundary condition?
- Page 8: What numerical method do you use for diffusion?
- Page 15 Line 54: You mention that $\sigma = 14 \mu\text{m}$. What does τ equal (ie what is the time step)?

COMMENTS ON FIGURES

- Figure 1: This figure is not necessary since the initial conditions are shown in Figure 4a. Or, it could be represented with both infected and healthy cells as red and green squares.
- Figure 4-5, 13-14: Need to include a color bar to define virus spread
- Figure 9: You may want to specify that green here represents infected cells (to match colors used in [29]) instead of healthy cells (like your other figures).

COMMENTS ON GRAMMAR/FORMATTING

- Page 3 Line 18, and Page 19 Line 49: Include reference. “HAL (Hybrid Automata Library) [14].”
- Page 3 Line 25-26: This sentence makes it sound like mathematical models have only been powerful tools since COVID19. Maybe rephrase to something like “across the globe, especially since the appearance of”

- Page 3 Line 36-37: “spread of SARS-CoV-2 and influenza infections on a cellular level.”
- Page 21 Line 55 and Page 22 Line 22: Should “supper” be “upper”?

Review form: Reviewer 2

Is the manuscript scientifically sound in its present form?

Yes

Are the interpretations and conclusions justified by the results?

Yes

Is the language acceptable?

Yes

Do you have any ethical concerns with this paper?

No

Have you any concerns about statistical analyses in this paper?

No

Recommendation?

Accept with minor revision (please list in comments)

Comments to the Author(s)

In the paper “A hybrid PDE-ABM model for viral dynamics with application to SARS-CoV-2 and influenza”, the authors develop and implement a novel hybrid continuum-discrete mathematical model in order to describe the spread of a virus into host cells.

The approach takes advantage of the separation in size between virus and host cells in order to describe the two subsystem with different mathematical formalisms. The virus is treated as a continuum variable (in both space and time), while host cells are described as discrete variable on a 2D grid that can switch between three states: healthy, infected, dead. These transitions are the result of the interactions and feedbacks between the continuum and discrete parts of the model.

Main advantage of this hybrid approach is that it retains advantages from both (here combined) methods: it is fast (as continuous approaches) and spatially detailed (as the ABM approaches). As a result, the hybrid method allows to monitor the dynamics of virus spread in the host both in space and time. Another advantage is that the ABM method brings randomness (stochastic probabilities for cell state transitions). The incorporation of stochastic terms nicely makes the model applicable to simulations of biological phenomena.

In the paper, this novel hybrid method is applied to the study of both SARS-CoV-2 and influenza, and the results present differences in time/space evolution of these two.

The paper is very interesting and the conclusions very well supported by the results. However, a main (minor) point of revision for the authors is addressing the lack of focus in the presentation of the paper. I strongly believe that this paper would largely benefit from a reorganization and more focused presentation.

Therefore, I recommend publication of this paper, pending minor revision.

Please consider the following suggestions:

-I find the introduction not focused: it describes how and why mathematical models are a good tool to study phenomena, but it goes "too far" with examples. I recommend just staying around a couple of examples of models for the viral transmission between cells and only mention those methods that are directly related to the ones used here=PDE and ABM.

- The descriptions of applications of ODE and ABM should be focused on a couple of examples of COVID and influenza only

-Novelty is stated very weakly, as "this hybrid approach has not been applied to explore the spatio-temporal dynamics of the virus". I think that the main point the authors should make needs to be related with the importance of monitoring in space and time how the virus system evolves, and what information we lose if we don't account for that in models.

-Innovation is also stated vaguely: applying this model allows to "examine several fundamental dynamic features of the virus in a new light". This sentence simply does not bring any information. what features and what light?

Several similar points should be addressed in the presentation of results and conclusion as well.

As I mentioned earlier, I find the model very well formulated and rigorous, and the application of this new mathematical method novel and interesting. The results are also comprehensive. The conclusions are well supported by the data. However, I recommend revisiting the overall presentation of the paper by focusing it more.

Decision letter (RSOS-210787.R0)

Dear Mr Marzban

The Editors assigned to your paper RSOS-210787 "A hybrid PDE-ABM model for viral dynamics with application to SARS-CoV-2 and influenza" have now received comments from reviewers and would like you to revise the paper in accordance with the reviewer comments and any comments from the Editors. Please note this decision does not guarantee eventual acceptance.

Please submit your revised manuscript and required files (see below) no later than 21 days from today's (ie 23-Jul-2021) date. Note: the ScholarOne system will 'lock' if submission of the revision is attempted 21 or more days after the deadline. If you do not think you will be able to meet this deadline please contact the editorial office immediately.

on behalf of Professor Mark Chaplain (Subject Editor)
openscience@royalsociety.org

Associate Editor Comments to Author:

The reviewers have a number of comments that the editors would like you to address, please. There several comments regarding the scientific aspects of the work (for example, definitions of terms etc), and there appears to be a need to re-work the paper's presentation to make it more accessible to a reader. Please can you take care to revise the paper taking these concerns into account - the editorial office will provide more details in a decision letter regarding the requirements for the revision.

Reviewer comments to Author:

Reviewer: 1
Comments to the Author(s)

SUMMARY

These authors create a hybrid agent based model to simulate influenza and SARS-CoV-2 and compare outputs to an ODE model. They claim that investigating the spatiotemporal dynamics provides new insights that are not captured with the ODE model. This manuscript was easy to read and I thought the approach was interesting. However, I thought there needed to be a little more discussion on why the results from the hybrid model are important and/or potentially useful for treatment (discussed below).

COMMENTS ON INTRODUCTION

- Page 4 Line 29-33: You claim that the “most urging current biological questions are already well-known.” Could you please clarify what these questions are, or what you mean by this?

COMMENTS ON RESULTS/DISCUSSION

- Page 12 Line 43-46: Define a specific time point at which the higher diffusion value results in a higher number of infected cells, since it isn't true for all time points.
- Page 12 Line 47-49: This paragraph needs more description related to what is different between Figures 6 and 7 (for example, timing and sharpness of peaks, magnitude, etc.).
- How much variability was there between simulations? For example, were there ever any cases where healthy cells did not die out because the infection went extinct?
- A paragraph or two is needed to describe why these results are important. In the abstract, the authors claim that “we can gain insight into why the outcomes of these two infections are

different.” However, the authors should describe more about what is known to be different between the two infections. For example, how do their R_0 's compare? How does the spatial model provide insight into the differences seen in Figure 12? Furthermore, the authors claim that “spatiotemporal information ... is vital from the aspect of disease treatment” but do not elaborate on how these results can help motivate potential treatment strategies. Please describe a bit more on this.

COMMENTS ON METHODS

- Page 8 Line 43-33: Describe why you use no-flux boundary conditions for the ABM. Do you expect results to be dependent on choice of boundary condition?
- Page 8: What numerical method do you use for diffusion?
- Page 15 Line 54: You mention that $\sigma = 14 \mu\text{m}$. What does τ equal (ie what is the time step)?

COMMENTS ON FIGURES

- Figure 1: This figure is not necessary since the initial conditions are shown in Figure 4a. Or, it could be represented with both infected and healthy cells as red and green squares.
- Figure 4-5, 13-14: Need to include a color bar to define virus spread
- Figure 9: You may want to specify that green here represents infected cells (to match colors used in [29]) instead of healthy cells (like your other figures).

COMMENTS ON GRAMMAR/FORMATTING

- Page 3 Line 18, and Page 19 Line 49: Include reference. “HAL (Hybrid Automata Library) [14].”
- Page 3 Line 25-26: This sentence makes it sound like mathematical models have only been powerful tools since COVID19. Maybe rephrase to something like “across the globe, especially since the appearance of”
- Page 3 Line 36-37: “spread of SARS-CoV-2 and influenza infections on a cellular level.”
- Page 21 Line 55 and Page 22 Line 22: Should “supper” be “upper”?

Reviewer: 2

Comments to the Author(s)

In the paper “A hybrid PDE-ABM model for viral dynamics with application to SARS-CoV-2 and influenza”, the authors develop and implement a novel hybrid continuum-discrete mathematical model in order to describe the spread of a virus into host cells.

The approach takes advantage of the separation in size between virus and host cells in order to describe the two subsystem with different mathematical formalisms. The virus is treated as a continuum variable (in both space and time), while host cells are described as discrete variable on a 2D grid that can switch between three states: healthy, infected, dead. These transitions are the result of the interactions and feedbacks between the continuum and discrete parts of the model.

Main advantage of this hybrid approach is that it retains advantages from both (here combined) methods: it is fast (as continuous approaches) and spatially detailed (as the ABM approaches). As a result, the hybrid method allows to monitor the dynamics of virus spread in the host both in space and time. Another advantage is that the ABM method brings randomness (stochastic probabilities for cell state transitions). The incorporation of stochastic terms nicely makes the model applicable to simulations of biological phenomena.

In the paper, this novel hybrid method is applied to the study of both SARS-CoV-2 and influenza, and the results present differences in time/space evolution of these two.

The paper is very interesting and the conclusions very well supported by the results. However, a main (minor) point of revision for the authors is addressing the lack of focus in the presentation of the paper. I strongly believe that this paper would largely benefit from a reorganization and more focused presentation.

Therefore, I recommend publication of this paper, pending minor revision.

Please consider the following suggestions:

-I find the introduction not focused: it describes how and why mathematical models are a good tool to study phenomena, but it goes "too far" with examples. I recommend just staying around a couple of examples of models for the viral transmission between cells and only mention those methods that are directly related to the ones used here=PDE and ABM.

- The descriptions of applications of ODE and ABM should be focused on a couple of examples of COVID and influenza only

-Novelty is stated very weakly, as "this hybrid approach has not been applied to explore the spatio-temporal dynamics of the virus". I think that the main point the authors should make needs to be related with the importance of monitoring in space and time how the virus system evolves, and what information we lose if we don't account for that in models.

-Innovation is also stated vaguely: applying this model allows to "examine several fundamental dynamic features of the virus in a new light". This sentence simply does not bring any information. what features and what light?

Several similar points should be addressed in the presentation of results and conclusion as well.

As I mentioned earlier, I find the model very well formulated and rigorous, and the application of this new mathematical method novel and interesting. The results are also comprehensive. The conclusions are well supported by the data. However, I recommend revisiting the overall presentation of the paper by focusing it more.

===PREPARING YOUR MANUSCRIPT===

If you have been asked to revise the written English in your submission as a condition of publication, you must do so, and you are expected to provide evidence that you have received

language editing support. The journal would prefer that you use a professional language editing service and provide a certificate of editing, but a signed letter from a colleague who is a native speaker of English is acceptable. Note the journal has arranged a number of discounts for authors using professional language editing services (<https://royalsociety.org/journals/authors/benefits/language-editing/>).

===PREPARING YOUR REVISION IN SCHOLARONE===

<https://royalsociety.org/journals/authors/author-guidelines/#supplementary-material> to

include a suitable title and informative caption. An example of appropriate titling and captioning may be found at https://figshare.com/articles/Table_S2_from_Is_there_a_trade-off_between_peak_performance_and_performance_breadth_across_temperatures_for_aerobic_sc_ope_in_teleost_fishes_/3843624.

Author's Response to Decision Letter for (RSOS-210787.R0)

See Appendix A.

RSOS-210787.R1 (Revision)

Review form: Reviewer 1

Is the manuscript scientifically sound in its present form?

Yes

Are the interpretations and conclusions justified by the results?

Yes

Is the language acceptable?

Yes

Do you have any ethical concerns with this paper?

No

Have you any concerns about statistical analyses in this paper?

No

Recommendation?

Accept with minor revision (please list in comments)

Comments to the Author(s)

See attached file (Appendix B).

Review form: Reviewer 2

Is the manuscript scientifically sound in its present form?

Yes

Are the interpretations and conclusions justified by the results?

Yes

Is the language acceptable?

Yes

Do you have any ethical concerns with this paper?

No

Have you any concerns about statistical analyses in this paper?

No

Recommendation?

Accept as is

Comments to the Author(s)

The authors have fully addressed my comments and suggestions

Decision letter (RSOS-210787.R1)

Dear Mr Marzban

On behalf of the Editors, we are pleased to inform you that your Manuscript RSOS-210787.R1 "A hybrid PDE-ABM model for viral dynamics with application to SARS-CoV-2 and influenza" has been accepted for publication in Royal Society Open Science subject to minor revision in accordance with the referees' reports. Please find the referees' comments along with any feedback from the Editors below my signature.

Please submit your revised manuscript and required files (see below) no later than 7 days from today's (ie 15-Sep-2021) date. Note: the ScholarOne system will 'lock' if submission of the revision is attempted 7 or more days after the deadline. If you do not think you will be able to meet this deadline please contact the editorial office immediately.

on behalf of Prof Mark Chaplain (Subject Editor)
openscience@royalsociety.org

Associate Editor Comments to Author:

Please see the comments attached from Reviewer 1 - you will need to address their comments regarding the results in your response to reviewers (and also the manuscript itself, of course).

Reviewer comments to Author:

Reviewer: 1

Comments to the Author(s)

See attached file

Reviewer: 2

Comments to the Author(s)

The authors have fully addressed my comments and suggestions

===PREPARING YOUR MANUSCRIPT===

===PREPARING YOUR REVISION IN SCHOLARONE===

Author's Response to Decision Letter for (RSOS-210787.R1)

See Appendix C.

Decision letter (RSOS-210787.R2)

Dear Mr Marzban,

I am pleased to inform you that your manuscript entitled "A hybrid PDE-ABM model for viral dynamics with application to SARS-CoV-2 and influenza" is now accepted for publication in Royal Society Open Science.

on behalf of Mark Chaplain (Subject Editor)
openscience@royalsociety.org

Appendix A

Response to reviewers

We are grateful to the Section Editor for handling our manuscript, and to the reviewers for their work and useful comments. We are delighted that both reviewers found our manuscript interesting, and we have implemented their suggestions in the revision. Our point-by-point response is below.

Reviewer 1

COMMENTS ON INTRODUCTION

- Page 4 Line 29-33: You claim that the “most urging current biological questions are already well-known.” Could you please clarify what these questions are, or what you mean by this?

Answer: The introduction has been streamlined and this sentence is not included any more.

COMMENTS ON RESULTS/DISCUSSION

- Page 12 Line 43-46: Define a specific time point at which the higher diffusion value results in a higher number of infected cells, since it isn't true for all time points.

Answer: Our word of choice was indeed unfortunate, but what we meant by “infected cells” in this particular sentence was the total number of cells that got infected throughout the simulation. In other words, we referred to the total number of “lost, damaged cells”, i.e. cells that are either currently infected now or are already dead. This has been made clear in the revised version. Some additional descriptions have also been added related to what is different between Figures 4 and 5.

- Page 12 Line 47-49: This paragraph needs more description related to what is different between Figures 6 and 7 (for example, timing and sharpness of peaks, magnitude, etc.).

Answer: Further details have been added in the corresponding paragraph related to what has been pointed out by the reviewer.

- How much variability was there between simulations? For example, were there ever any cases where healthy cells did not die out because the infection went extinct?

Answer: Assuming the configuration we set in our article, the chance of the infection spontaneously vanishing is negligible. In our simulations we typically considered ~20 initially infected cells out of the total of 40.000 cells; R_0 is approximately 12,05 for SARS-CoV-2 -- according to our results virus spread always thrives in this particular setting. In order to illustrate the stochastic variability of our model for the parameters we use in our article (those described in Figure 6, specifically), we ran the simulation with the same setting a hundred times. Our results are given in Figure R1 -- the solid lines and the shaded areas correspond to the mean and standard deviation, respectively.

However, these parameters can be manipulated to facilitate spontaneous virus extinction. For example, by increasing virus clearance and contemporarily decreasing the initial number of infected cells, we can obtain a scenario where randomness is significant in the final output, as Figure R2. illustrates. This image shows the number of infected cells in two different simulations using the exact same parameter values (see caption for details) and initial conditions $I_0= 2$, $H_0= 40000-I_0$, $V_0= 0$ and $D_0= 0$. The increased virus clearance can be a surrogate of a positive immune response or an effect of treatment for example.

The discussion of such stochastic variability of outputs would be interesting and we may consider this in more detail in a follow-up work.

Figure R1: The output of the hybrid PDE-ABM system using the parameters given in (3.2) and a diffusion coefficient value of 0,2. The results represent the corresponding mean and standard deviation values obtained by 100 different stochastic simulations.

Figure R2: The number of infected cells in two different simulations using the exact same parameters. The stochastic nature of the hybrid PDE-ABM system allows nondeterministic variability in the results. (The figures were obtained by using the parameters given in (3.2), a diffusion coefficient value of 0,2; and the increased viral removal rate = $9 \cdot 10^{-2}$)

- A paragraph or two is needed to describe why these results are important. In the abstract, the authors claim that “we can gain insight into why the outcomes of these two infections are different.” However, the authors should describe more about what is known to be different between the two infections. For example, how do their R_0 's compare? How does the spatial model provide insight into the differences seen in Figure 12?

Answer:

1. We have added more details at the discussion of Figure 12, and we have noted right after the respective paragraph that the difference between the two images of Figure 12 will be addressed in even more detail after some important observations are made in connection to Figures 13 and 14 first.

2. As the reviewer suggests, we have added new paragraphs after the original discussion of Figures 13 and 14 -- these new paragraphs refer back to Figure 12 in a new light. In this new part we emphasise the advantage and importance of the hybrid model and point out virological differences between the two viruses that are made visible by the hybrid model's results.

3. In particular, the definition of R_0 itself -- especially within the context of the hybrid model -- is not necessarily identical to the well-known formula expressed for the ODE system, and we do not delve into this topic within the scope of the present article. We do refer however to differences resulting from features such as different death probability values (connection to real data), which do affect the reproduction number.

Furthermore, the authors claim that “spatiotemporal information ... is vital from the aspect of disease treatment” but do not elaborate on how these results can help motivate potential treatment strategies. Please describe a bit more on this.

Answer: The key in applying our proposed model to evaluating potential drug effectiveness lies in its ability to capture diffusion. Spatial heterogeneity is vital -- the penultimate paragraph (dedicated to this topic) in the Discussion section has been updated with this information.

COMMENTS ON METHODS

- Page 8 Line 43-33: Describe why you use no-flux boundary conditions for the ABM. Do you expect results to be dependent on choice of boundary condition?

Answer: The respective paragraph has been updated, where we provide justification for the choice and usefulness of Neumann boundary conditions.

- Page 8: What numerical method do you use for diffusion?

Answer: We apply the implicit method implemented internally by HAL [10] -- the complete HAL library is available at <https://github.com/MathOnco/HAL>, their

respective *PDEGrid2D.java* implements the DiffusionADI function, which, as its name suggests, uses the alternating direction implicit method. Our code (<https://github.com/sadeghmarzban/Hybrid-PDE-ABM-for-viral-dynamics>) calls this specific DiffusionADI function. The information regarding using the implicit method has been added in a new paragraph within the Implementation section.

- Page 15 Line 54: You mention that $\sigma=14\mu\text{m}$. What does τ equal (i.e. what is the time step)?

Answer:

We begin with an update regarding sigma itself. Since diffusion is given as “ $0.19 \mu\text{m}^2 / \text{min}$ ”, and on the level of HAL the value of 0.19 is interpreted as “unit space² / unit time” (i.e. cell / unit time), for simplicity we choose sigma = 1 μm . The respective part containing 14 μm has been removed. The time step is 1 minute in the specific applications, this information has been added to both Table 2 and Table 3 below each table. Note that the referenced part has been moved to Appendix A.

COMMENTS ON FIGURES

- Figure 1: This figure is not necessary since the initial conditions are shown in Figure 4a. Or, it could be represented with both infected and healthy cells as red and green squares.

Answer: The respective roles of Figure 1 and Figure 4/a are different. While Figure 1 is more of a “sketch” dedicated to give a better and clearer idea about the model itself and how its implementation works, Figure 4. presents the output of an actual simulation. In some sense Figure 1. is a blueprint, and Figure 4. is the result. Also, Figure 4/a does not show the individual cells very clearly. Hence, we feel it is best to keep both images, but we have changed the colours of Figure 1. -- in particular, healthy cells are now green as suggested by the reviewer.

- Figure 4-5, 13-14: Need to include a color bar to define virus spread

Answer: The color bars have been added.

- Figure 9: You may want to specify that green here represents infected cells (to match colors used in [29]) instead of healthy cells (like your other figures).

Answer: The following sentence has been added in the caption of Figure 9a.: Note the colour choice we apply in this figure: in order to match our simulation’s colours to the experimental results in \cite{Outlawe01935-20}, in this particular image the colour green represents viral particles rather than healthy cells.

COMMENTS ON GRAMMAR/FORMATTING

- Page 3 Line 18, and Page 19 Line 49: Include reference. “HAL (Hybrid Automata Library) [14].”

Answer: A reference has been added.

- Page 3 Line 25-26: This sentence makes it sound like mathematical models have only been powerful tools since COVID19. Maybe rephrase to something like “across the globe, especially since the appearance of”

Answer: This sentence has been rephrased.

• Page 3 Line 36-37: "spread of SARS-CoV-2 and influenza infections on a cellular level."
Answer: This specific text has been erased due to other modifications requested by the second Reviewer.

• Page 21 Line 55 and Page 22 Line 22: Should "supper" be "upper"?
Answer: The typos have been corrected in the revised manuscript.

Reviewer

2

• I find the introduction not focused: it describes how and why mathematical models are a good tool to study phenomena, but it goes "too far" with examples. I recommend just staying around a couple of examples of models for the viral transmission between cells and only mention those methods that are directly related to the ones used here=PDE and ABM.

Answer: To make it more focused, we have streamlined the Introduction to which has also been reorganised: reoccurring information has been eliminated -- essentially the entire first paragraph has been removed as it contained a lot of information that was stated again later at some point, other paragraphs have been correspondingly updated.

A new paragraph has been created dedicated to state novelty in a clear way.

• The descriptions of applications of ODE and ABM should be focused on a couple of examples of COVID and influenza only.

Answer:

Non- COVID / influenza related examples have been removed, and consequently the references that became irrelevant have been removed as well.

The PDE example has been updated to (sars-cov-2--odes-pdes), a newly added source dedicated to SARS-CoV-2-related research and includes within-host investigations as well.

We have added the host-level result of (ABM-influenza-hostlevel) to replace between-host examples; similarly, the more general (hybrid-multi-scale-abm) has been added to replace non-epidemiological references.

• Novelty is stated very weakly, as "this hybrid approach has not been applied to explore the spatio-temporal dynamics of the virus". I think that the main point the authors should make needs to be related with the importance of monitoring in space and time how the virus system evolves, and what information we lose if we don't account for that in models.

Answer: The last paragraph of the Introduction is a new paragraph dedicated to state the novelties.

• Innovation is also stated vaguely: applying this model allows to "examine several fundamental dynamic features of the virus in a new light". This sentence simply does not bring any information. what features and what light?

Answer: The particular sentence has been removed, and the novelty aspects are explained in more detail.

• Several similar points should be addressed in the presentation of results and conclusion as well.

Answer:

1. There is an overlap between this request and that pointed by the first reviewer, e.g. the focused, detail-oriented, improved description of several figures:
- In the Results section we have added details in connection with Figures 4 and 5; Figures 6 and 7; and Figures 12, 13 and 14 as well.
- In the Discussion we have also added a detailed description about why our model could be of great use in the evaluation process of potential treatment options.
2. To enhance focus and readability, we have moved two technical paragraphs from the Results section to the new "Appendix A", the latter is called "Technical details related to parameter values".

Additional changes:

1. We have updated our description on the initial number of infected cells. We introduce χ (on page 9) describing the number of infected cells at $t = 0$ and we use this χ in Table 3.
2. On page 15 we have added the image from [22] assessing in vitro viral spread, as [22] is an open-access article distributed under the terms of the Creative Commons Attribution 4.0 International license. The image is included in Figure 9b.
3. Occurrences of $V_{\{\Omega_{i,j}\}}$ has been corrected to $V^h(\Omega_{i,j})$ throughout the article as the hybrid virus concentration function is denoted by V^h , not V .
4. We have added a more detailed description of the relationship between the parameters P_I and β (page 6-7).
5. A missing " $\cdot \tau$ " has been added in the P_I term in the "Stochastic implementation of a healthy cell's infection" on page 10.
6. Minor grammatical changes have been made in Figure 2.

Appendix B

RESPONSE

- I appreciate the thorough response to the reviewer's comments, and am happy with the changes that the authors made to the manuscript. I only have one remaining comment about the unit of diffusion in the hybrid ABM (as well as minor comments on grammar/formatting).

COMMENTS ON METHODS

- Page 24 Table 3: You mention that you use $\sigma=1 \mu\text{m}$ instead of $\sigma=14 \mu\text{m}$ for technical simplicity (Appendix A). However, it is easy (and the convention) to implement the more biologically relevant units by setting the unit space in the ABM as $14 \mu\text{m}$ and converting the diffusion coefficient to reflect this, i.e. $0.19 \mu\text{m}^2\text{min}^{-1} = 0.19/(14^2) \text{units}^2\text{min}^{-1} = 0.00097 \text{units}^2\text{min}^{-1}$. Before acceptance, I would recommend that the authors check that this does not dramatically change results for Figures 9-14.

COMMENTS ON GRAMMAR/FORMATting

- Page 12 Line 20: "at formulated above" should be "as formulated above"
- Page 12 Line 30: "effects" should be "affects"
- Page 14 Line 55 & 57: 40.000 should be written as 40,000 to be consistent with the format throughout the rest of the manuscript
- Page 20 Line 15: $D_{V1}=0,2$ should be written as 0.2 to be consistent with the format throughout the rest of the manuscript
- Page 24 Line 50 & 51: 0,5 should be 0.5; 4,8 should be 4.8 to be consistent with the format throughout the rest of the manuscript
- Page 24 Table 3: cell^{-1} should be $[\text{infected cell}]^{-1}$

Appendix C

Dear Section Editor,

We were delighted that our Manuscript RSOS-210787.R1 "A hybrid PDE–ABM model for viral dynamics with application to SARS–CoV–2 and influenza" has been accepted for publication in Royal Society Open Science subject to minor revision.

Accordingly, we corrected the typos as suggested by Reviewer 1, and also followed their recommendation by applying the unit conversion for the diffusion coefficient (added to Appendix A) and using coherent diffusion coefficients in the text and figures of Section 3 (b). The details are below.

Reviewer 1:

“COMMENTS ON METHODS: Page 24 Table 3: You mention that you use $\sigma=1 \mu\text{m}$ instead of $\sigma=14 \mu\text{m}$ for technical simplicity (Appendix A). However, it is easy (and the convention) to implement the more biologically relevant units by setting the unit space in the ABM as $14 \mu\text{m}$ and converting the diffusion coefficient to reflect this, i.e. $0.19 \mu\text{m}^2\text{min}^{-1} = 0.19/(14^2) \text{units}^2 \text{min}^{-1} = 0.00097 \text{units}^2\text{min}^{-1}$. Before acceptance, I would recommend that the authors check that this does not dramatically change results for Figures 9-14.”

Answer: the respective diffusion values have been updated coherently in Table 3 with further clarifications in Appendix A.

Respective changes in Table 3:

- We have set $\sigma = 14\mu\text{m}$.
- The diffusion values used in HAL are now adjusted to have units σ^2 / min

Respective changes in the main body of the article:

- Due to the similar size of SARS-CoV-2 and influenza virus particles, we opt for a coherent $0.2 \sigma^2 / \text{min}$ choice, chosen from a reasonable range supported by the literature.
- Any parts that were related to this issue have been updated, see the coloured version.
- Figures of Section 3(b) were also updated accordingly, with additional figures to explore a wider range of parameters added to our github repository, which is referenced in the text

Respective changes in Appendix A:

- We have added a part dedicated to diffusion coefficient - related issues: we elaborate unit conversion and discuss the values from the literature.

COMMENTS ON GRAMMAR/FORMATting:

Page 12 Line 20: “at formulated above” should be “as formulated above” **Done.**

Page 12 Line 30: “effects” should be “affects” **Done.**

Page 14 Line 55 & 57: 40.000 should be written as 40,000 to be consistent with the format throughout the rest of the manuscript. **Done.**

Page 20 Line 15: DV1=0,2 should be written as 0.2 to be consistent with the format throughout the rest of the manuscript. **Done. (“0.2” had an additional occurrence on Page 12, it has been updated to “0,2” as well)**

Page 24 Line 50 & 51: 0,5 should be 0.5; 4,8 should be 4.8 to be consistent with the format throughout the rest of the manuscript. **Done.**

Page 24 Table 3: cell-1 should be [infected cell]-1. **Done.**

Reviewer 2 had no further comments and suggestions for Revision 1.

Sincerely,

[authors]